# Integrative network analysis of early-stage lung adenocarcinoma identifies aurora kinase inhibition as interceptor of invasion and progression

Seungyeul Yoo [1,2,3,16], Abhilasha Sinha[4,5,16], Dawei Yang[4,5,6], Nasser K. Altorki [7], Radhika Tandon [8], Wenhui Wang[1,2], Deebly Chavez[4], Eunjee Lee[1,2,3], Ayushi S. Patel [4,5,9], Takashi Sato [4,5,10,11], Ranran Kong[4,5,12], Bisen Ding[4,13], Eric E. Schadt[1,2,3,5], Hideo Watanabe [1,2,4,5], Pierre P. Massion[14,17], Alain C. Borczuk [15], Jun Zhu [1,2,3,5,18✉] & Charles A. Powell [4,5,18✉]

Here we focus on the molecular characterization of clinically significant histological subtypes of early-stage lung adenocarcinoma (esLUAD), which is the most common histological subtype of lung cancer. Within lung adenocarcinoma, histology is heterogeneous and associated with tumor invasion and diverse clinical outcomes. We present a gene signature distinguishing invasive and non-invasive tumors among esLUAD. Using the gene signatures, we estimate an Invasiveness Score that is strongly associated with survival of esLUAD patients in multiple independent cohorts and with the invasiveness phenotype in lung cancer cell lines. Regulatory network analysis identifies aurora kinase as one of master regulators of the gene signature and the perturbation of aurora kinases in vitro and in a murine model of invasive lung adenocarcinoma reduces tumor invasion. Our study reveals aurora kinases as a therapeutic target for treatment of early-stage invasive lung adenocarcinoma.

[1] Department of Genetics and Genomic Sciences, Icahn School of Medicine at Mount Sinai, New York, NY, USA. [2] Icahn Institute for Data Science and Genomic Technology, New York, NY, USA. [3] Sema4, Stamford, CT, USA. [4] Division of Pulmonary, Critical Care and Sleep Medicine, Icahn School of Medicine at Mount Sinai, New York, NY, USA. [5] Tisch Cancer Institute, Icahn School of Medicine at Mount Sinai, New York, NY, USA. [6] Department of Pulmonary and Critical Care Medicine, Zhongshan Hospital, Fudan University, Shanghai, China. [7] Department of Cardiothoracic Surgery, Weill Cornell Medicine-New York Presbyterian Hospital, New York, NY, USA. [8] School of Medicine, St. George's University, West Indies, Grenada. [9] Vileck Institute of Graduate Biomedical Sciences, New York University School of Medicine, New York, NY, USA. [10] Division of Pulmonary Medicine, Department of Medicine, Keio University School of Medicine, Tokyo, Japan. [11] Department of Respiratory Medicine, Kitasato University School of Medicine, Sagamihara, Japan. [12] Department of Thoracic Surgery, The Second Affiliated Hospital of Medical School, Xi'an Jiaotong University, Xi'an, Shaanxi, China. [13] Key Laboratory of Birth Defects and Related Diseases of Women And Children of MOE, State Key Laboratory of Biotherapy, West China Second University Hospital, Sichuan University, Chengdu, Sichuan, China. [14] Department of Medicine, Vanderbilt University Medical Center, Nashville, TN, USA. [15] Department of Pathology, Weill Cornell Medicine, New York, NY, USA. [16]These authors contributed equally: Seungyeul Yoo, Abhilasha Sinha. [17]Deceased: Pierre P. Massion. [18]These authors jointly supervised this work: Jun Zhu, Charles A. Powell. ✉email: jun.zhu@mssm.edu; charles.powell@mssm.edu

Lung cancer has the second highest cancer incidence and is the leading cause of cancer-related mortality worldwide with annually 2.1 million new lung cancer cases and 1.8 million deaths[1]. Over the past 30 years, the overall five-year survival for lung cancer in the United States has increased from 12% to over 19% and continues to rise[2]. The increase is attributable to the implementation of early detection screening programs and the development of targeted therapies and immunotherapies that are effective in specific subtypes of metastatic lung cancer. These advances have been essential for the improved outcomes for lung cancer. In this study, we present an approach that focuses on therapeutic approaches that target molecular subtypes of early-stage lung adenocarcinoma (esLUAD) and promise to improve outcomes for this important patient group.

Lung adenocarcinoma is the most common histological subtype of lung cancer. Although esLUAD patients have much better prognosis than patients with advanced disease, among early-stage patients treated primarily with surgery, about 30–50% of early-stage patients will develop metastasis with 70% and 35% overall 5-year survival rate for stage 1 and 2 non-small cell lung cancer patients, respectively[3,4]. Within lung adenocarcinoma, histology is heterogeneous and associated with diverse tumor invasion and clinical outcomes[5]. Invasiveness is one of the cancer hallmarks and is directly related to metastatic potential and clinical outcomes of the tumor[6].

Yu and colleagues examined integrated gene/protein expression data of resected lung adenocarcinomas from The Cancer Genome Atlas LUAD cohort (TCGA) and showed that the integrated genomic models improved survival prediction of stage 1 tumors[7]. However, the analysis cannot be extrapolated to examine the impact of histological invasiveness because the TCGA LUAD cohort does not include non-invasive histological subtypes such as minimally invasive adenocarcinoma (MIA) or adenocarcinoma in situ (AIS). Recently, Xing and colleagues profiled indolent LUAD subsolid nodules and tumor micro-environment and showed a unique transcriptomic signature involving cell–cell interaction pathways that was distinct from primary LUAD with lymph node metastasis[8]. Our previous studies reported the type 2 TGF Beta Receptor (*TGFβRII*) as a significant determinant of invasiveness and metastasis of localized lung adenocarcinoma[4,9,10]. While these previous studies were useful to confirm the molecular and clinical heterogeneity in the esLUAD, the identification of treatment opportunities guided by molecular subtypes of early lung cancer remains underdeveloped[11–13].

Here, we show invasiveness mechanisms in esLUAD by analyzing gene expression of a cohort of 53 histologically heterogeneous esLUAD samples. Based on the transcriptomic analysis of this cohort, we identify a gene signature that distinguishes invasive and non-invasive tumors and validated the prognostic significance in multiple independent cohorts. We then perform a systematic analysis to understand the molecular functions of the signature genes. Integrative network analysis highlights *TPX2*, (an activator/binding partner of Aurora kinase A[14,15]), and Aurora kinase B (*AURKB)* as key regulators of the pro-invasive signature. Aurora kinases are a family of highly conserved serine-threonine kinases and are master regulators of mitotic cell division, chromosomal segregation and spindle formation[16,17]. Among the three members of aurora kinase family (A, -B and -C), Aurora kinase A and B are the best characterized. Their role in tumor growth and survival has been reported across several cancer types[18,19], but the association of aurora kinases with lung tumor invasion or its role in early lung cancer therapy is unclear. In this work, we define the role of aurora kinases in early lung cancer progression and demonstrate using in vitro and in vivo models that targeting this pathway reduces lung cancer invasiveness and improves survival.

## Results

**Lung adenocarcinoma invasiveness signature**. We performed molecular profiling of 53 histologically heterogeneous esLUAD samples (Supplementary Table 1) by RNA sequencing and identified signature genes associated with invasiveness of tumors at an early stage (Methods, Supplementary Note 1). First, we performed an unsupervised hierarchical clustering for the 53 samples based on the most varying genes (SFig. 1a). The unsupervised hierarchical clustering results in two distinct groups containing 20 and 33 samples, respectively (SFig. 1a). Differentially expressed genes (DEGs) between the two groups were determined based on *t*-test using cutoffs (fold-change (FC) > 1.5 and FDR < 0.01) and the samples were re-clustered into two groups of 21 and 32 tumors, respectively, based on the DEGs (Fig. 1a). Samples in each group are consistent with their histological subtypes; Group 1 on the left side (red color in Inv. Class) is enriched for pathological aggressive subtypes acinar (AC), micro papillary (MP), papillary (PAP), and solid (SOL) while Group 2 on the right side (blue color in non-Inv. Class) consists of less invasive or non-invasive tumors such as MIA, AIS, and lepidic predominant (LPA) (Fig. 1a). Therefore, we annotate Group 1 as "Invasive" and Group 2 as "Non-invasive" (Supplementary Table 1). Together, 1322 DEGs that distinguish the Invasive and Non-invasive tumors are identified as invasiveness signature genes (Supplementary Data 1). These invasiveness signature genes are separated into 526 genes upregulated in the Invasive group (pro-invasive signature genes) and 796 genes upregulated in the Non-invasive group (indolence signature genes). EGFR mutation is enriched within the non-invasive tumors classified by the signature (chi-square test $p = 0.003$, SFig. 1b) while KRAS and TP53 mutation are observed in both invasive and non-invasive tumors (chi-square test $p = 0.33$ and 0.07 for KRAS and TP53, respectively, SFig. 1b). The signature is not associated with sex or smoking status (Supplementary Note 2).

The clustering of 53 tumors aligns with histological subtypes and staging of the samples (Chi-square test $p = 1.4 \times 10^{-7}$ and 0.001 for histology and nodal stage, respectively, Fig. 1b). The Invasive group is highly enriched for tumors with aggressive histology (AC, MP, PAP, and SOL, 17 out of 20, Fisher's exact test (FET) OddRatio (OR) = 36.42 and $p = 1.5 \times 10^{-7}$) while the Non-Invasive group includes less invasive histological tumors (MIA, AIS, and LPA, 29 out of 32, FET OR = 50.17 and $p = 2.1 \times 10^{-8}$). Most node-positive tumors are included in the Invasive group (9 Out of 10, FET OR = 21.8 $p = 0.0005$).

To compare our signature genes derived from clustering with those derived from tumor classification, we identified 793 histology-based DEGs between aggressive tumors (AC, MP, PAP, and SOL) and others (MIA, AIS, and LPA) (SFig. 1c, left) and 80 nodal-based (N.stage) DEGs between node-positive and -negative tumors (SFig. 1c, right). Our signature genes capture larger molecular differences between invasive and non-invasive tumors than histology- and node stage-based signatures containing 92% and 89% of histology-based and N.stage-based signatures, respectively (Fig. 1c). Compared to histology-based signatures, the expression differences of the overlapping genes between the signatures by the two approaches are more significant in the expression-driven approach (SFig. 1d), suggesting that tumor classification by gene expression-driven clustering is more robust for comparing invasiveness phenotypes among esLUAD tumors than tumor classification by histology or node metastasis status alone.

**The signature genes are enriched for tumor invasion-related functions**. We functionally annotated the pro-invasive and

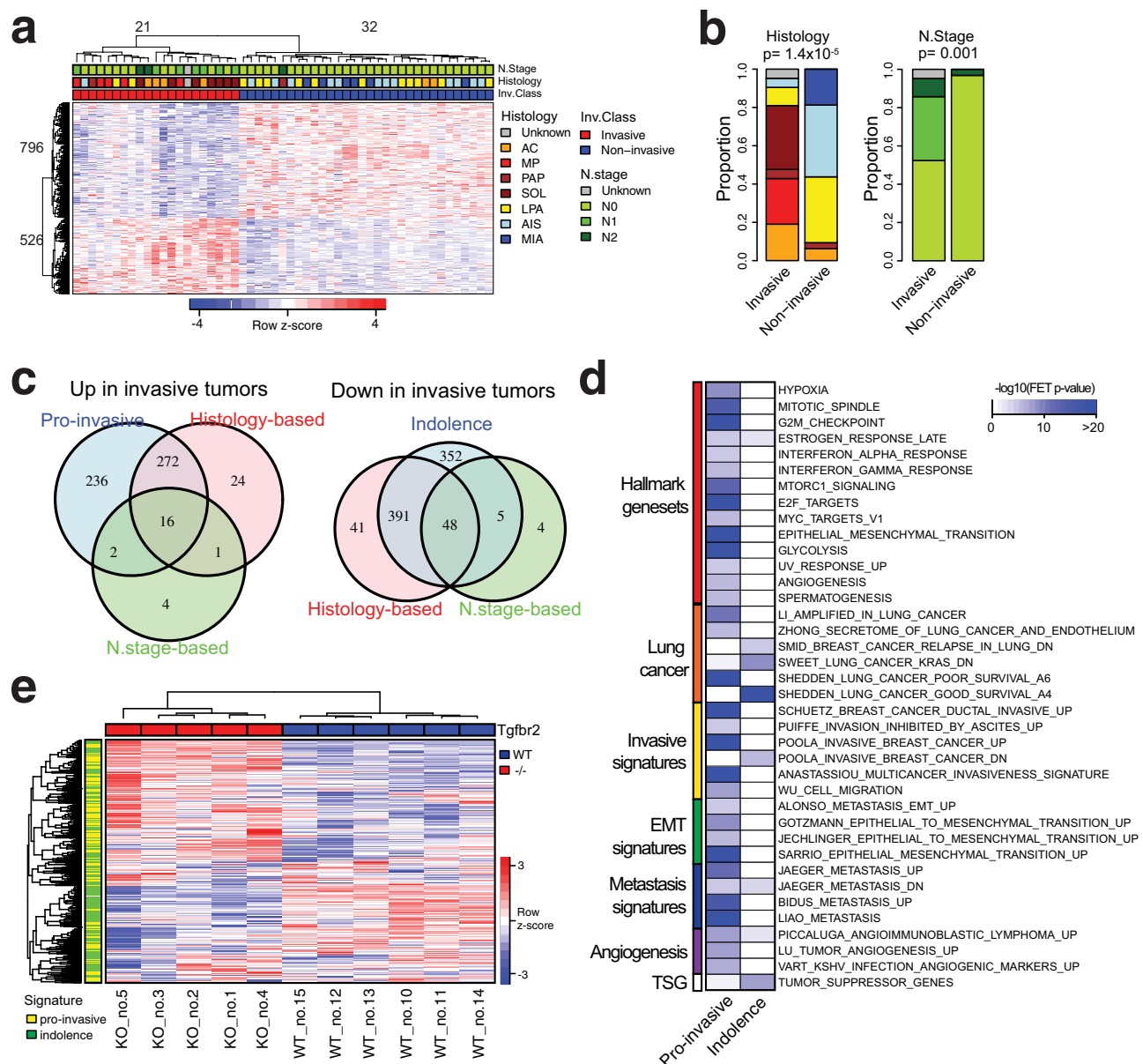

**Fig. 1 Pro-invasive and indolence signature genes associated with tumor invasion at early-stage lung adenocarcinoma. a** Invasiveness signature genes separating 53 early-stage lung adenocarcinoma into invasive and non-invasive tumors. "Complete" method was used for unsupervised clustering. **b** Comparison of the tumor clusters based on the signature genes with histological subtypes and nodal stages of the tumors. *P*-values of association are measured by a two-tailed Chi-Square test. Color codes for histology and nodal stages are same as panel (**a**). **c** Comparison of the gene numbers of pro-invasive and indolence signatures with differentially expressed genes based on histological subtypes (AIS, MIA, LPA vs AC, MP, PAP, SOL) and nodal stage (N0 vs N1+). Lists of genes in each category are provided in Source Data files. **d** Association of pro-invasive and indolence signatures with MSigDB gene sets including Hallmark, lung cancer-related genes, tumor invasion, or metastasis-related gene sets. **e** Unsupervised clustering of 11 tumors (5 Tgfbr2 −/− and 6 Tgfbr2 wt mice from GSE27717) based on the signature genes. "Complete" method was used for unsupervised clustering. Expression profiles of the signature genes are provided in Source Data files.

indolence signatures using public databases (Methods). First, gene ontology (GO) terms curated in the MSigDB C5 collection[20] were compared with our signature genes. The pro-invasive signature genes are significantly enriched for the cell cycle (FET OR = 5.1, $p = 2.9 \times 10^{-42}$), cell division (FET OR = 6.9, $p = 5.2 \times 10^{-31}$), extracellular matrix (FET OR = 5.9, $4.6 \times 10^{-22}$), kinetochore (FET OR = 11.9, $p = 1.3 \times 10^{-19}$), and other cell cycle-related functions. The indolence signature genes are enriched for terminally differentiated cell activities such as extracellular matrix (FET OR = 3.0, $p = 2.4 \times 10^{-9}$), tissue development (FET OR = 1.8, $p = 1.8 \times 10^{-7}$), and plasma membrane region (FET OR =

2.1, $p = 3.7 \times 10^{-7}$). The complete list of GO terms enriched in the pro-invasive and indolence signatures is shown in Supplementary Data 2. Compared with the Hallmark gene sets[21], the pro-invasive signature genes significantly overlap not only with cell cycle associated gene sets such as mitotic spindle, G2M checkpoint, or E2F target genes, but also with EMT and angiogenesis pathways which are major hallmarks of tumor invasion and metastasis functions[6] (Fig. 1d).

Next, we compared the signatures with previously reported gene sets related to lung cancer and to tumor invasive properties (MSigDB C2 collection) (Fig. 1d). The pro-invasive and indolence

signatures significantly overlap with poor and good survival lung cancer genes reported by Shedden et al.[22] (FET OR = 20.7 and 7.6, $p = 8.7 \times 10^{-110}$ and $8.4 \times 10^{-23}$, respectively), and they also significantly overlap with upregulated and downregulated in invasive breast cancer[23] (FET OR = 14.4 and 3.9, $p = 3.2 \times 10^{-51}$ and $3.1 \times 10^{-6}$, respectively). The pro-invasive signature genes are also consistently enriched for other signatures associated with EMT[24–26], metastasis[27–29], and angiogenesis[26,30,31] (Supplementary Data 3). Interestingly, the indolence signature significantly overlap with known tumor suppressor genes curated in TSGene2.0[32] (FET OR = 2.1, $p = 3.5 \times 10^{-8}$). These results suggest that our signature genes derived from lung adenocarcinoma are strongly associated with biological pathways important for tumor progression and are generalizable to invasive and/or metastatic features of epithelial neoplasms.

Tumor proliferation is a fundamental component of tumor malignancy and a well-known phenotype contributing to patients' prognosis[33], thus it is expected that cell cycle genes will be represented in our invasiveness signature. The pro-invasive signature genes include 57 of 131 genes in the tumor proliferation signature meta-PCNA gene set[34] (FET OR = 36.8 and $p = 5.3 \times 10^{-57}$). We examined the impact of proliferation gene expression by clustering the 53 LUAD tumors using only the meta-PCNA signature genes; 10 of invasive tumors are clustered together with non-invasive tumors (SFig. 1e, left). However, when we use our signature genes, excluding the meta-PCNA signature genes, the clustering result completely agrees with the original clustering (SFig. 1e, right), suggesting that tumor proliferation alone is not sufficient to explain the molecular differences between invasive and non-invasive tumors and that our signature genes capture tumor invasive specific biologic features in addition to tumor proliferation features.

We compared functional enrichment of the expression-driven clustering signatures with those of histology-based DEGs (Supplementary Note 3). Although both gene signatures are enriched for multiple common MsigDB gene sets including cell cycle-related gene sets, more significant enrichment for tumor invasion-related gene sets is generally observed with the pro-invasive signature genes (SFig. 1f). Moreover, genes uniquely identified in the pro-invasive signature are significantly enriched for tumor invasion-related gene sets such as MULTICANCER_INVASIVE_SIGNATURE (FET OR = 27.25 and $p = 9.9 \times 10^{-17}$) and EMT (FET OR = 10.9 and $p = 1.4 \times 10^{-17}$) (SFig. 1g) suggesting that the expression-driven clustering approach not only increases the number of genes of interest but also captures unique molecular features related to tumor invasiveness compared to histology- or node metastasis status-based classification.

**The signature genes are associated with the invasive phenotype of Kras$^{+/-}$Tgfbr2$^{-/-}$ mouse model.** Previously, we showed the loss of *Tgfbr2* in the non-invasive murine inducible lung adenocarcinoma model (Kras$^{+/-}$Tgfbr2$^{-/-}$) induces a highly invasive phenotype associated with lymph node metastasis and poor survival[35]. *TGFBR2* is included in the human lung adenocarcinoma indolence signature (FDR = 0.0016, Supplementary Data 1). We examined the overlay of human lung adenocarcinoma gene signatures with murine gene expression from invasive (Tgfbr2$^{-/-}$) and non-invasive (Tgfbr2$^{wt}$) tumors and show that the pro-invasive and indolence signatures classify tumors from 11 mice into two groups (Methods) that exactly match their *Tgfbr2* status (Fig. 1e). The Tgfbr2$^{-/-}$ mice (KO) have higher expression of pro-invasive signatures while Tgfbr2$^{wt}$ (WT) show opposite patterns (Fig. 1e), confirming the significant role of *Tgfbr2* in transforming non-invasive tumors into invasive tumors. Comparing our human esLUAD signatures with murine KO vs WT DEGs (Methods), the pro-invasive signatures strongly overlap

with genes upregulated in KO (FET OR = 5.0, $p = 1.6 \times 10^{-12}$) while the indolence genes are enriched for genes upregulated in WT tumors (FET OR = 2.1, $p = 0.002$) (Supplementary Table 2).

**The signature genes are associated with early-stage patients' survival.** Because signature genes of invasiveness correspond to a critically important biological function of tumor progression that is associated with clinical outcomes (Fig. 1d, e), we hypothesized that our gene signatures could predict survival of esLUAD patients examined by independent studies. Using the pro-invasive and indolence signature genes, we developed an invasiveness score (IVS) based on the elastic net to estimate the tumor's invasiveness (Methods). Gene expression profiles of seven independent lung adenocarcinoma cohorts were downloaded, and stage I and II tumors were selected for further analysis (Methods). Based on the IVS, samples were classified as "Indolent" (peaked at 0) or "Invasive" (peaked at 1) and samples with IVS value between the two peaks were classified as "Intermediate" (SFig. 2a, Methods). For each independent lung cancer cohort, we calculated IVS of each sample and associated the scores with patient survival. IVS captures a clear transition of expression of gene signatures of tumor samples in each of 7 independent datasets (SFig. 2b). In all seven independent datasets, "Invasive" groups have significantly inferior survival compared to "Indolent" groups (log-rank test (LRT) $p = 8.3 \times 10^{-5}$, 0.003, 0.0002, 0.0007, 0.0003, 0.02, and 0.05 for Shedden, TCGA, Okayama, Tang, Der, Rousseaux, and Wilkerson datasets, respectively, Fig. 2a–g). To assess the robustness of the survival association of IVS-based classification, we used multiple parametric and non-parametric IVS cutoff schemes to classify tumors. Both fixed IVS cutoff value and fixed IVS percentile-based classifications show segregation of patient groups with significantly different survival (Supplementary Table 3). The hazard ratios (HRs) between the high vs. low-risk groups based on the dataset specific cutoff (Methods) are generally larger than corresponding HRs based on splits into three groups of equal size (SFig. 3, Supplementary Table 3). With adjustment for age, sex, and tumor stages, the IVS remains significantly associated with survival in these cohorts (SFig. 2c).

Considering the heterogeneity of populations from the international datasets, the strong and consistent detection of survival differences highlights the biological significance of the invasive signature genes as important prognostic biomarkers for early lung adenocarcinoma patient survival. Notably, most of the international datasets such as TCGA do not contain MIA or AIS tumors, yet our invasiveness signature genes distinguish patients with different survival, suggesting that our signature genes are generalizable for understanding biological and clinical pathways that determine survival outcomes in esLUAD.

**The signature genes rank invasiveness phenotype of cancer cell lines.** To evaluate invasiveness signature mechanisms in vitro, we calculated IVS as relative invasiveness in 70 lung adenocarcinoma cell lines from the CCLE dataset[36]. The 70 cell lines are ranked based on the IVS (Fig. 2h). To validate the biological implications of the IVS-based ranking, we experimentally assessed cell invasiveness using in vitro migration and invasion assays. We selected eight high IVS (ranked above median) with five KRAS mutant: H1373, SK-LU-1, H1792, A549, and H2009; three KRAS wild type: H1755, H1650, and H1975; and five low IVS (ranked below median) HCC78, H3255, Calu3, HCC1833, HCC2279 cell lines. All eight high IVS cell lines show higher transwell migration and invasion within 48 h time period independent of KRAS status compared to low IVS score cell lines (t-test p-values = $6.5 \times 10^{-21}$ and $8.7 \times 10^{-24}$, respectively, Fig. 2i, j for migration and 2k, l for

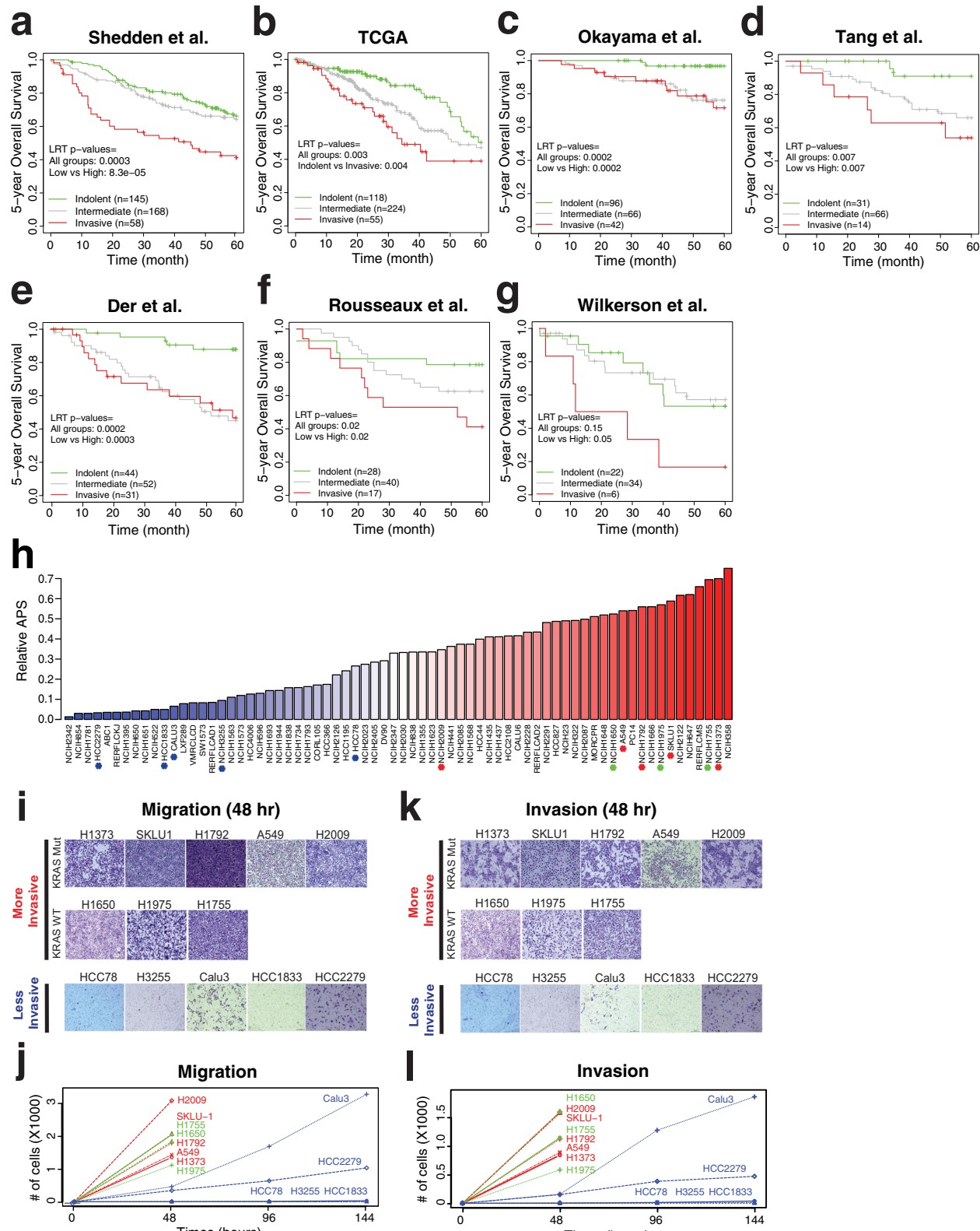

invasion), suggesting that the invasiveness phenotype predicted by our IVS-based ranking is significantly associated with experimentally observed biological phenotypes and that KRAS mutation status is not a determining factor of invasiveness in these cells. Furthermore, the less invasive lines HCC1833, HCC78, and H3255 do not show significant migration or invasion until 144 h of assay (Fig. 2j, l and SFig. 4a). Calu3 and HCC2279 have low migration and invasion at 48 h compared to more invasive cells (*t*-test *p*-values = $9.7 \times 10^{-18}$ and $3.7 \times 10^{-22}$ in migration and invasion, respectively). Although all LUAD cell

**Fig. 2 Invasiveness score associated with patient survival and tumor cells phenotype. a** Stratification of tumor samples of Shedden et al. cohort[22] into three groups; invasive (high IVS), intermediate (middle IVS), and indolent (low IVS) tumors determined based on IVS (Methods, SFig. 2a). Five-year survival of tumors is shown in a KM curve with corresponding LRT p-values. Source data are provided in Source Data files. **b** KM plot for TCGA LUAD cohort[76]. **c** KM plot for Okayama et al. cohort[77]. **d** KM plot for Tang et al. cohort[79]. **e** KM plot for Der et al. cohort[78]. **f** KM plot for Rousseaux et al. cohort[80]. **g** KM plot for Wilkerson et al. cohort[81]. **h** Relative Adenocarcinoma Progression Score (APS, values 0–1) of 70 LUAD cell lines estimated via IVS. Tumor cells above the median score are labeled as "More Invasive" (red color bars) and below as "Less Invasive" (blue color bars). In each group, we selected a few cell lines for further experimental validations as marked in asterisk (red for more invasive cells with KRAS mutation, green for more invasive ones with wildtype KRAS, and blue for less invasive cells). Source data are provided in Source Data files. **i** Representative images of migrated cells in transwell migration assay after 48 h in eight more invasive and five less invasive LUAD cells. Scale bar 10 μm. **j** Quantification of migrated cells through transwell at 48, 96, and 144 h. Data presented as mean (n = 3). Source data are provided in Source Data files. Red: more invasive Kras mutant; Green: more invasive Kras WT; Blue: less invasive. **k** Representative images of invaded cells in transwell matrigel invasion assay after 48 h in eight more invasive and five less invasive LUAD cells. Scale bar 10 μm. **l** Quantification of invaded cells through transwell matrigel at 48, 96, and 144 h. Data presented as mean (n = 3). Source data are provided in Source Data files. Red: more invasive Kras mutant; Green: more invasive Kras WT; Blue: less invasive.

lines were derived from invasive LUAD tumors, the results above show that the invasiveness phenotype varies across the LUAD cell lines and that our gene signatures enable us to infer a relative invasive phenotype. To determine the potential effect of proliferation rate on the invasiveness of these cell lines, we performed a growth rate assay to examine differences in growth rates between the cell lines predicted as more invasive versus less invasive. Using seven more invasive cells (H1792, A549, H1975, H2009, H1650, H1373, and SK-LU-1) and five less invasive cells (H3255, HCC78, Calu3, HCC1833, and HCC2279), we observed no significant difference of growth rates between the two groups (t-test $p = 0.13$, 0.46, 0.86, and 0.75 for day 2, 3, 4 and 5, respectively, SFig. 4b). This suggests that the invasiveness activity of LUAD cells is independent of basal proliferation rates.

**Integrative network and connectivity map analysis identifies key drivers and therapeutic targets in early lung adenocarcinoma.** The lung adenocarcinoma invasiveness signature distinguishes tumors of different histology and more importantly, of distinct clinical outcomes following surgical resection. The biological mechanisms that drive the transformation of tumors from indolence to invasiveness remain incompletely characterized. Hence, we constructed a molecular regulatory network for esLUAD by integrating gene expression, CNV, and methylation profiles of the stage I LUAD samples in TCGA using the software package RIMBANET (Methods)[37,38]. The final network consists of 8,533 genes including 389 pro-invasive signature genes and 562 indolence signature genes (SFig. 5). Applying key driver analysis (Methods), we identify 13 and 9 key drivers regulating the pro-invasive and indolence signature genes, respectively (Supplementary Table 4). The pro-invasive signature genes form two large closed subnetworks consisting of 123 and 97 nodes, respectively. The first subnetwork is regulated by *TPX2*, *AURKB*, and five other key regulators (referred as TPX2/AURKB subnetwork in Fig. 3a), and the second subnetwork is driven by collagen associated genes *COL1A2* and *COL11A1*, and 3 other key regulators (referred as COL1A2 subnetwork in Fig. 3b). *TPX2* is a well-known activator of *AURKA* and vice versa and thus together regulate spindle assembly and dynamics[39,40]. *AURKB* is a master regulator of mitosis that forms the enzymatic core of the Chromosomal Passenger Complex (CPC) with Survivin (*BIRC5*)[41,42]. Both *AURKA* and *BIRC5* are also in the TPX2/AURKB subnetwork (Fig. 3a).

We interrogated molecular functions enriched within these subnetworks. Although there are distinct molecular pathways associated with each subnetwork (Supplementary Table 5), i.e., the TPX2/AURKB network is enriched for cell cycle-related genes including G2M_CHECKPOINT and E2F_TARGETS (FET OR = 66.93 and 56.4 with $p = 1.45 \times 10^{-59}$ and $8.71 \times 10^{-57}$, respectively) and the COL1A2 network is enriched for HYPOXIA

and ANGIOGENESIS (FET OR = 8.07 and 25.16 with $p = 1.83 \times 10^{-7}$ and $5.68 \times 10^{-7}$, respectively) in HALLMARK datasets, both subnetworks are enriched in gene sets such as HALLMARK MTORC1 signaling pathways (FET OR = 4.9 and 4.93 with $p = 9.37 \times 10^{-5}$ and 0.0004 for the TPX2/AURKB and COL1A2 subnetworks, respectively) and invasive signatures in breast cancer[23] (FET OR = 18.68 and 7.42 with $p = 5.7 \times 10^{-29}$ and $1.93 \times 10^{-8}$, respectively). In addition, genes in the TPX2/AURKB subnetwork are enriched for up-regulated genes in EMT in breast cancer[26] (FET OR = 53.12 and $p = 1.02 \times 10^{-51}$) and genes in the COL1A2 subnetwork are associated with HALLMARK EMT pathway genes (FET OR = 33.16 and $p = 1.85 \times 10^{-34}$) (Supplementary Table 5). These results suggest that even though the two subnetworks of the pro-invasive signatures are not directly connected with each other, both subnetworks share common downstream pathways associated with tumor invasiveness functions.

To identify drugs that could impact tumor invasiveness and potentially prevent tumor metastasis, we searched the Connectivity map (CMAP) database to identify small molecules that could perturb the two pro-invasive subnetworks[43] (Methods). A list of potential perturbagens for each subnetwork was identified based on CMAP enrichment scores from 2429 small molecule compounds in A549 cells. Aurora kinase inhibitor is among the top 10 perturbagens for both subnetworks (enrichment score < −95, Fig. 3c). When considering the two TPX2/AURKB and COL1A2 subnetworks together, the strongest perturbagen class to reverse the expression of the genes in the two subnetworks is aurora kinase inhibitor (Fig. 3c). Taken together, the genomic analyses suggest that aurora kinases represent potential regulators of invasiveness that are vulnerable to small molecular inhibitors.

**Aurora kinases A and B (Aurora-A and Aurora-B) protein expression in human lung adenocarcinoma tumors is associated with pathological invasiveness and survival.** To examine the clinical significance of Aurora-A and Aurora-B in esLUAD, we examined protein expression in a large series of resected human tumors represented on a tissue microarray (TMA) (Methods, Supplementary Table 6). Categorizing tumors into non-invasive and invasive groups based on their histology (Non-invasive_histology: MIA, AIS, LPA vs. Invasive_histology: AC, PAP, MP, SOL), patients in the Invasive_histology group show significantly worse survival than those in Non-invasive_histology group (LRT $p = 0.001$, SFig. 6a). Aurora-A and Aurora-B expression are significantly correlated with each other at the protein ($p = 1.1 \times 10^{-5}$) level, similar to the correlation at the mRNA level ($p = 6.7 \times 10^{-14}$ and $1.3 \times 10^{-14}$ noted in the original and TCGA datasets, respectively) (SFig. 6b). The Aurora-A and Aurora-B immunostaining scores are significantly higher in the Invasive_histology group (t-test $p = 2.0 \times 10^{-12}$ and

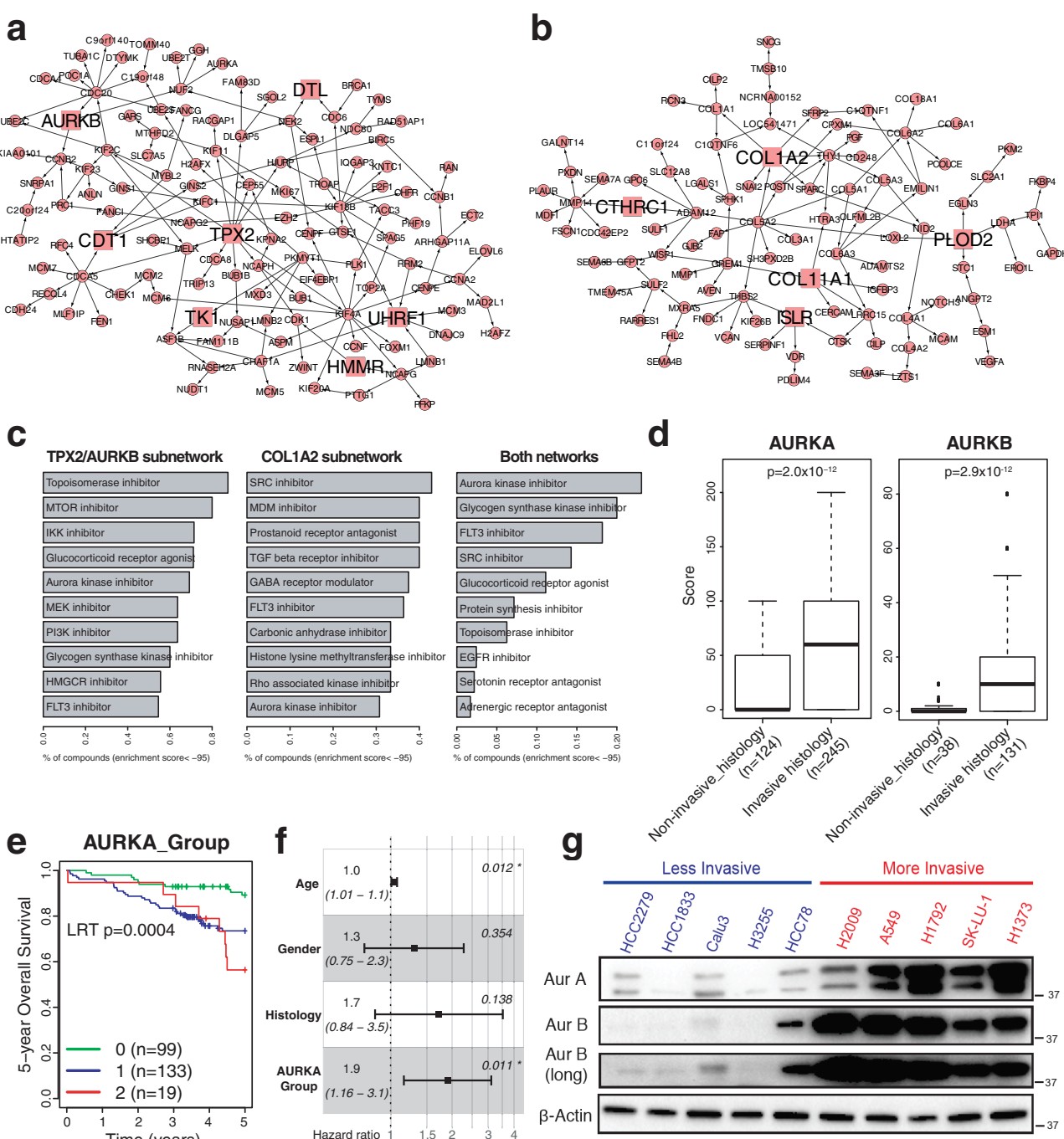

**Fig. 3 AURKA and AURKB are putative master regulators regulating pro-invasive signatures. a** A subnetwork of 123 pro-invasive genes driven by *TPX2* and *AURKB*. Seven putative master regulators are marked by squares with larger size. Source data are provided in Source Data files. **b** A subnetwork of 97 pro-invasive genes driven by *COL1A2*. Five putative master regulators are marked by squares with larger size. Source data are provided in Source Data files. **c** Top 10 small molecular classes significantly reversing expression of genes in the two subnetworks in A549 using CMAP database (enrichment score < −95, Methods): left for TPX2/AURKB subnetwork, middle for COL1A2 subnetwork, and right for both. Source data are provided in Source Data files. **d** Comparison of Aurora-A and B protein scores between non-invasive_histology (AIS, MIA, LPA) and invasive_histology (AC, MP, PAP, SOL) tumors from human tissue microarray data. Aurora-B score are not available for all patients. The middle line of the box is median values for each group and the box edges are the 25th and 75th percentiles. Whiskers indicate maximum and minimum values of each group except for outliers. Two-side *t*-test *p*-values are shown. Source data are provided in Source Data files. **e** KM curve showing the survival differences between patients classified into 0, 1, 2+ based on Aurora-A score and the percentage of positive cells (Methods). The LRT p-value is estimated. Source data are provided in Source Data files. **f** Hazard ratios and 95% confidence intervals of multiple factors are measured by multivariate cox regression from Survival ~ age + sex + histology + AURKA_Group. Center dots indicate Hazard ratios and error bars indicate upper and lower 95% confidence intervals. Significant *p*-values are marked by asterisk. Source data are provided in Source Data files. **g** Western blot of Aurora-A and Aurora-B for panel of five less invasive and five more invasive LUAD cell lines from Fig. 2i. Source data are provided in Source Data files.

$2.9 \times 10^{-12}$, respectively, Fig. 3d, IHC in SFig. 6c, d), consistent with the observations based on mRNA expression data (SFig. 6e, f). Tumor clusters based on the Aurora-A score show different survival outcomes indicating the prognostic significance of Aurora-A (LRT $p$-value = 0.0004, Fig. 3e). Multivariate Cox-regression analysis show the survival association of Aurora-A is independent of age, sex, and histology of patients (hazard ratio = 1.9 with 95% confidence interval = [1.16, 3.1], $p$ = 0.01, Fig. 3f). In addition, we observed higher levels of Aurora-A and Aurora-B protein expression in the more invasive CCLE cells lines as compared to the less invasive cell lines (Fig. 3g).

**Aurora kinase A and B regulate migration and invasion phenotypes in lung adenocarcinoma cells.** Aurora kinase A and B are aberrantly expressed in several tumor types[44–49], and promote cancer proliferation and survival owing to their functions in cell cycle and mitosis[50]. However, their role in regulating the invasiveness and migratory phenotypes in human tumors is not well characterized. We used genetic and small molecule inhibition approaches to examine the effect of Aurora-A and/or Aurora-B on tumor cell migration and invasion. We performed CRISPR/Cas9 deletion of *AURKA* and *AURKB* in highly invasive H1792 and A549 cells (Methods, Fig. 4a, b, SFig. 7a and 8a). Repressing *AURKA* or *AURKB* alone does not alter migration and invasion in either cell line (Fig. 4c, d and SFig. 7b, c) but, deleting both *AURKA* and *AURKB* together demonstrates significant reduction in migration by 60% and 50% and invasion by 70% and 60% in H1792 and A549 cells, respectively at 48 h, adjusting for cell number (one-way ANOVA for all comparisons $p < 0.0001$ Fig. 4c, d and SFig. 8b, c). These results suggest that Aurora-A and Aurora-B have redundant functions in regulating lung tumor invasiveness and pan-inhibition is required to impact tumor cell invasion. To examine whether the reduction in migration and invasion by CRISPR/Cas9 knockout of *AURKA* and *AURKB* is influenced by an effect on cell proliferation, we performed a proliferation assay for H1792 and A549 cells from Day 1 to Day 8. We observed no significant difference in proliferation at 48 h after gene knockout, which is the time point that migration and invasion were measured (SFig. 8d). As expected, we observed a difference in cell proliferation in knockout cells at later time points. Our proliferation assay data indicate that the effect seen in migration and invasion assay is not influenced by gene deletion effect on cell proliferation at the 48 h time point.

To confirm the long-term viability of Aurora-A and Aurora-B knockout cells, we measured viability for CRISPR/Cas9 knockout of *AURKA* and *AURKB* cells at 8 days (SFig. 8e, f) and used fluorescence imaging to confirm that Aurora-A and Aurora-B knockout cells are viable during the assessment of both viability at early and later time points (SFig. 8g) and invasion and migration at day 2 (SFig. 8h).

**Small molecule pan-aurora kinase inhibitors suppress the activity of aurora kinases and invasive phenotype of lung adenocarcinoma cells.** Because our genetic experiments indicate co-operative regulation of lung cancer cell invasion by Aurora-A and Aurora-B, we focused small-molecule inhibitor studies on pan-aurora kinase inhibitors, AMG900 and PF-03814735. AMG900 is in phase 1 clinical trials for acute myeloid leukemia[51] (NCT01380756) and advanced solid tumors[52] (NCT00858377) and in phase 2 clinical trials for triple-negative breast cancer in combination with angiogenic kinase inhibitor ENMD-2076[53] (NCT01639248). PF-03814735 is in phase 1 clinical trial for advanced solid tumors[54] (NCT00424632). We treated highly invasive LUAD cell lines (H1792, A549, and H2009) with AMG900 or PF-03814735 for 48 h.

To determine the functional consequences of inhibiting Aurora-A and Aurora-B activity, we examined cell migration and invasiveness in five highly invasive cell lines (H1373, SK-LU-1, H1792, A549 and H2009, Fig. 2h–l) by treatment with AMG900 and PF-03814735. Both pan-aurora kinase inhibitors significantly suppress transwell migration and invasion assays in all 5 cell lines treated with either drug (Fig. 4e, SFig. 9a, and statistical test results in Supplementary Table 7). We also show a significant decrease in cell motility in wound healing assay except for H1373 with PF-03814735 treatment (Fig. 4e, SFig. 9a, and Supplementary Table 7). Migration and invasion are similarly reduced in invasive cells of wildtype KRAS (H1650 and H1975) after AMG900 treatment (two-way ANOVA test $p < 0.0001$ for dose of 0.1 and 1 µM, SFig. 9b) suggesting that pan-aurora kinase inhibition suppresses invasiveness in lung adenocarcinoma cells independent of KRAS driver mutation status. Cell line growth rates can affect the readout of migration and invasion assays. When migration and invasion assays were performed at 24 h, which is shorter than the doubling intervals of A549 and H2009 cells, we observed a similar impact by aurora kinase inhibition as in experiments performed at 48 h (SFig. 9c). This suggests that invasion and migration are regulated by aurora kinase signaling independent of effects on proliferation. We did not observe any effect on cell viability at 48 h in any of the LUAD cells treated in a concentration range of 0.001–10 µM (SFig. 9d).

**Aurora kinases inhibition decreases invasive signature gene expression and impairs AKT/mTOR and EMT signaling pathway.** Genetic and pharmacological inhibition of Aurora-A and Aurora-B impairs cell migration and invasiveness in lung adenocarcinoma cell lines (Fig. 4). To comprehensively understand the molecular mechanisms driven by inhibition of aurora kinases, we examined gene expression profiles of A549 and H1792 cells treated with an aurora kinase inhibitor (AMG900) and identified DEGs (FDR < 0.01) (Fig. 5a, SFig. 10a, and Supplementary Data 4). DEGs are highly consistent in both cell lines (FET OR = 10.5 and 14.2 and $p$-value = $10^{-1054}$ and $10^{-1572}$ for upregulated and downregulated genes, respectively) suggesting common effects of AMG900 treatment. They are enriched for pathways such as MTORC1_SIGNALING set (FET OR = 4.3 and 5.8 $p = 2.5 \times 10^{-20}$ and $2.6 \times 10^{-27}$ for A549 and H1792 cells, respectively, Fig. 5b and SFig. 10b, Supplementary Table 8). Genes downregulated by AMG900 treatment in A549 cells are significantly enriched for pro-invasive signature genes (FET OR = 5.6 and $p = 7.7 \times 10^{-62}$) and more significantly enriched for genes in the TPX2/AURKB subnetwork (FET OR = 123.14 and $p = 1.9 \times 10^{-104}$, Fig. 5c). Similar observations were made for H1792 cells (FET OR = 2.0 and 15.0 and $p = 7.2 \times 10^{-10}$ and $5.1 \times 10^{-16}$ for the pro-invasive signature and TPX2/AURKB genes, respectively, SFig. 10c). This indicates that AMG900 treatment transcriptionally perturbs the TPX2/AURKB subnetwork. Even though the COL1A2 subnetwork is not significantly downregulated in A549 and H1792 treated cells (FET $p = 0.61$ and 0.02), genes associated with the MTORC1 signaling pathway such as *PLOD2* and *LHDA* as well as an EMT gene, *SNAI2*, are suppressed by aurora kinase inhibition (Fig. 5c and SFig. 10c).

It has been shown that activation of p-S6-kinase and p-4E-BP1 pathways by mTORC1 promotes cell motility and invasion[55], but there are no data showing a relationship between aurora kinases with AKT/mTOR signaling and tumor cell invasiveness. Consistent with the RNAseq transcriptomic analysis, we observe decreased p-AKT (Thr308) and p-mTOR (Ser2448) protein expression in A549 and H1792 cells treated with AMG900 (Fig. 5d, SFig. 10d). Downstream targets of mTOR pathway activation, p-S6-kinase (Ser371) and p-4E-BP1 (Thr37/46) are

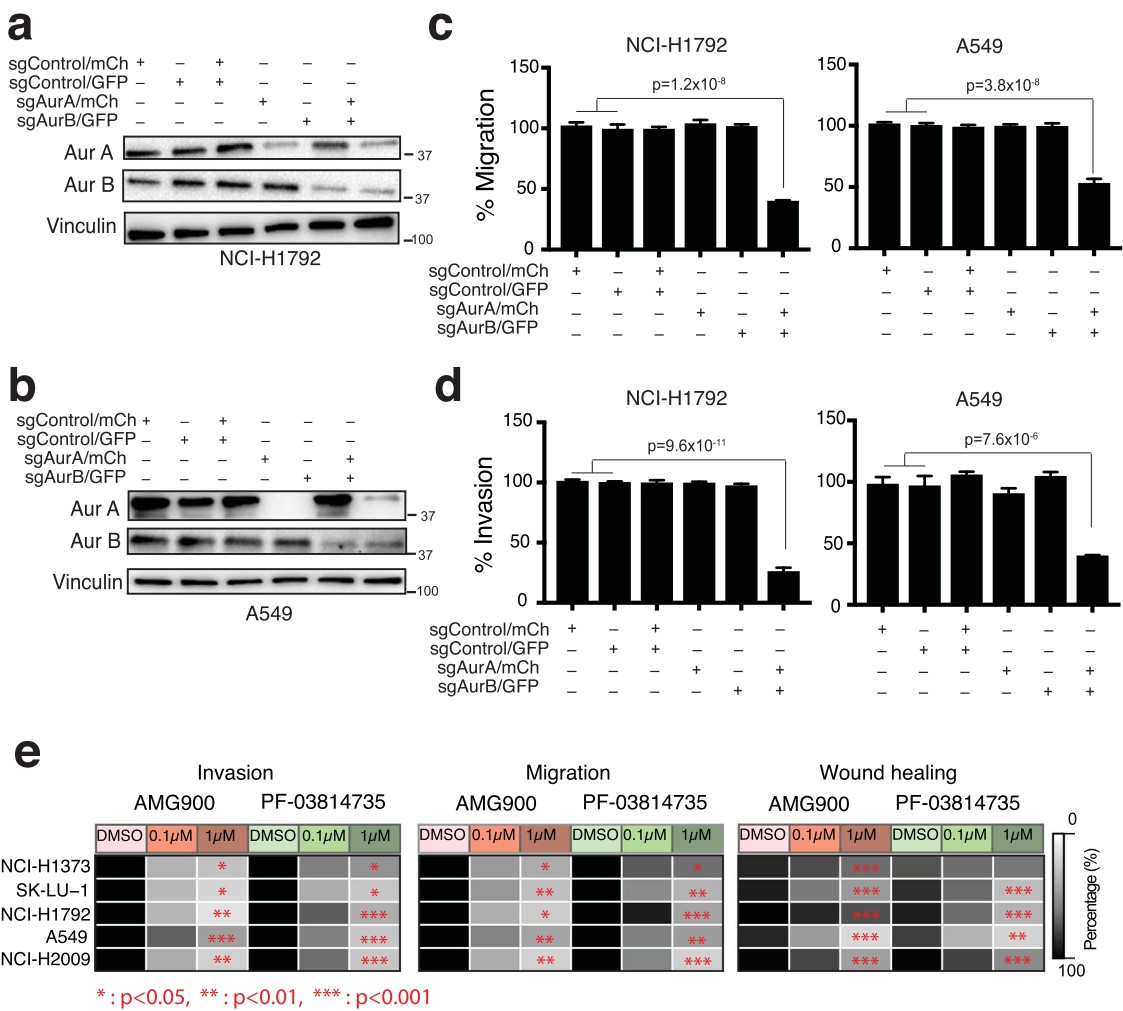

**Fig. 4 Aurora kinase A and B are both regulators of invasive phenotype. a** Western blot for H1792 cells transduced with indicated sgRNAs. Source data are provided in Source Data files. **b** Western blot for A549 cells transduced with indicated sgRNAs. Source data are provided in Source Data files. **c** Quantitation of percent migration of H1792 and A549 cells transduced with indicated sgRNAs at 48 hr. Data presented as mean ± s.e.m. one-way ANOVA $n = 4$, ****$p < 0.0001$. Source data are provided in Source Data files. **d** Quantitation of percent invasion of H1792 and A549 cells transduced with indicated sgRNAs at 48 h. Data presented as mean ± s.e.m. one-way ANOVA $n = 4$, ****$p < 0.0001$. Source data are provided in Source Data files. **e** Heat map for H1373, SK-LU-1, H1792, A549, and H2009 showing the effect of AMG900 and PF-03814735 on invasion and migration phenotype, and wound healing ability, at indicated drug concentrations ($n = 4$ for migration and invasion assay and $n = 8$ for wound healing assay). Two-side test $p$-values are measured using a generalized linear model (Methods). All statistical test results are listed in Supplementary Table 7. Source data are provided in Source Data files.

similarly suppressed. Aurora kinase inhibitor also suppresses ERK1/2 activity in both cell lines (Fig. 5d, SFig. 10d). The lung adenocarcinoma pro-invasive signature is enriched for EMT signature genes (Fig. 1d) that are suppressed in AMG900 treated H1792 cells (SFig. 10b, FET OR = 3.0, $p = 3.5 \times 10^{-10}$). In cells treated with aurora kinase inhibitor AMG900, aurora kinase inhibition downregulates mesenchymal markers N-Cadherin and Vimentin in A549 and H1792 (Fig. 5d, SFig. 10d) and upregulates E- Cadherin in A549. EMT markers such as Snail, Slug, and Claudin1 are also suppressed by aurora kinase inhibition, suggesting that aurora kinases mediate invasiveness and cell motility by regulating EMT pathways. To further examine the effect of *AURKA* and *AURKB* knockout on AKT/mTOR and EMT signaling, we examined protein expression for both pathways in CRISPR knockout A549 (Fig. 5e) and H1792 (SFig. 10e) cells. p-mTOR and p-AKT are suppressed in cells deleted for both *AURKA* and *AURKB*, but not with single deletion. Similarly, EMT marker N-Cadherin, Claudin 1, Vimentin, and Snail are suppressed only in cells with double deletion of *AURKA* and *AURKB* (Fig. 5e and SFig. 10e). Together,

these data suggest that AKT/mTOR and EMT pathway activation in invasive lung adenocarcinoma cells is dependent upon aurora kinase signaling.

The genes upregulated by AMG900 treatment do not overlap with the indolence signature genes in either cell line (FET $p = 0.32$ and 0.98 for A549 and H1792 cells, respectively) suggesting that molecular mechanisms underlying the indolence signature genes regulation are independent of aurora kinase activity.

**AMG900 intercepts lung adenocarcinoma invasiveness in Kras(G12D)/Tgfbr2$^{-/-}$ genetically engineered mouse model.** To investigate the pre-clinical utility of aurora kinase inhibitors and to evaluate its efficacy in suppressing invasive lung adenocarcinoma, we treated the transgenic invasive LUAD mouse model of KrasG12D mutation and inducible Tgfbr2 deletion with AMG900. We previously showed that loss of Tgfbr2 in Kras G12D mutant condition induces highly invasive phenotype[35] in the transgenic mouse model. We imaged tumor formation

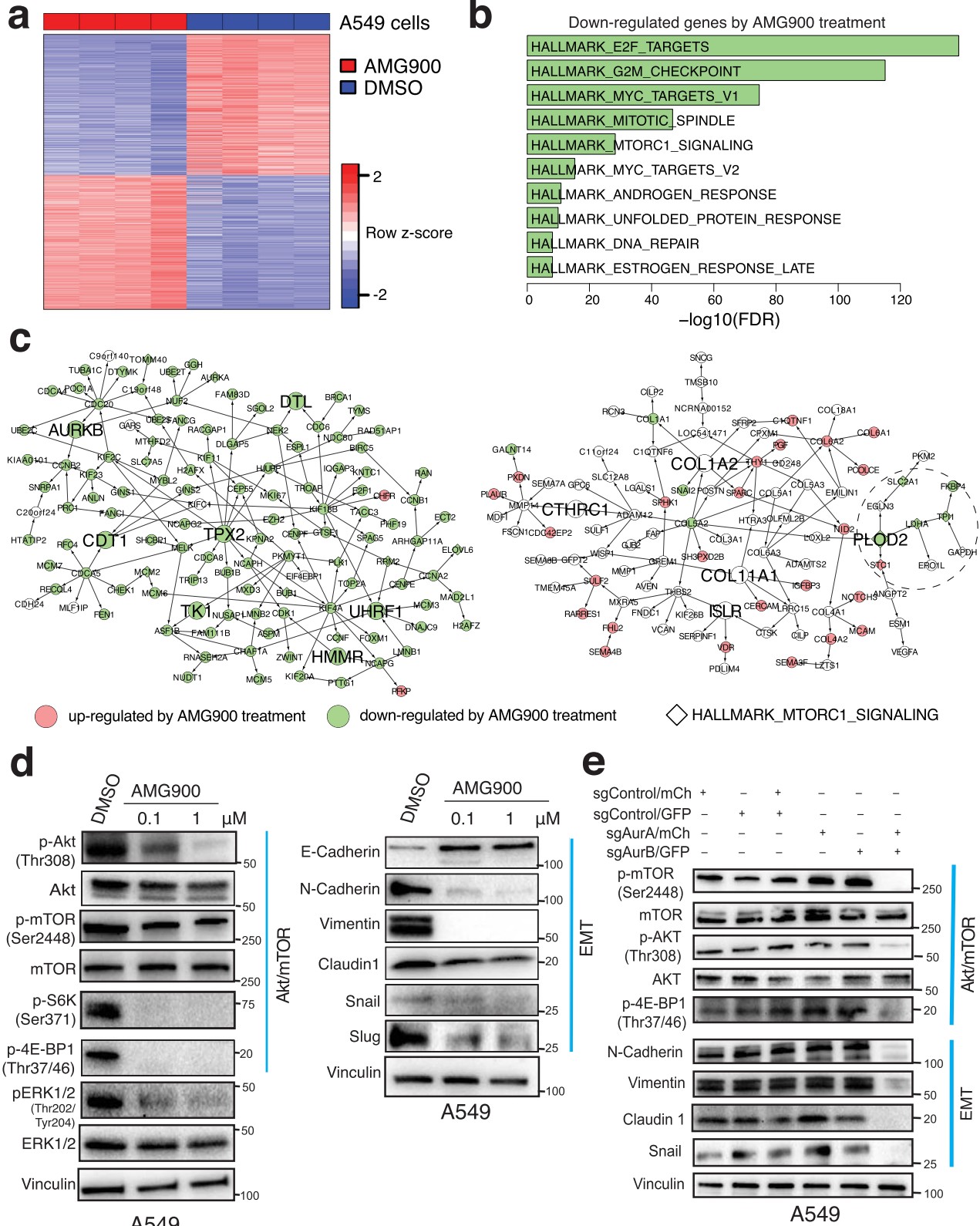

**Fig. 5 Aurora kinases drive invasiveness in lung adenocarcinoma through activating AKT/mTOR and EMT pathways. a** Differentially expressed genes between DMSO and 0.1 μM AMG900 treated A549 cells (FDR < 0.01). DEGs are listed in Supplementary Data 4. **b** Top 10 downregulated HALLMARK pathways in A549 treated with 0.1 μM AMG900. **c** Overlaying the DEGs in A549 cells onto the TPX2/AURKB and COL1A2 subnetworks. Nodes filled in red are upregulated and ones in green are downregulated by AMG900 treatment. Genes included in HALLMARK_MTORC1_SIGLANING pathways are indicated with diamonds (◇) within a dashed circle. **d** Western blot for A549 cells treated with DMSO and indicated concentrations of AMG900 for 48 h for AKT/mTOR pathway and EMT pathway. Source data are provided in Source Data files. **e** Western blots of markers of AKT/mTOR and EMT pathways in A549 cells treated with indicated sgRNAs. Source data are provided in Source Data files.

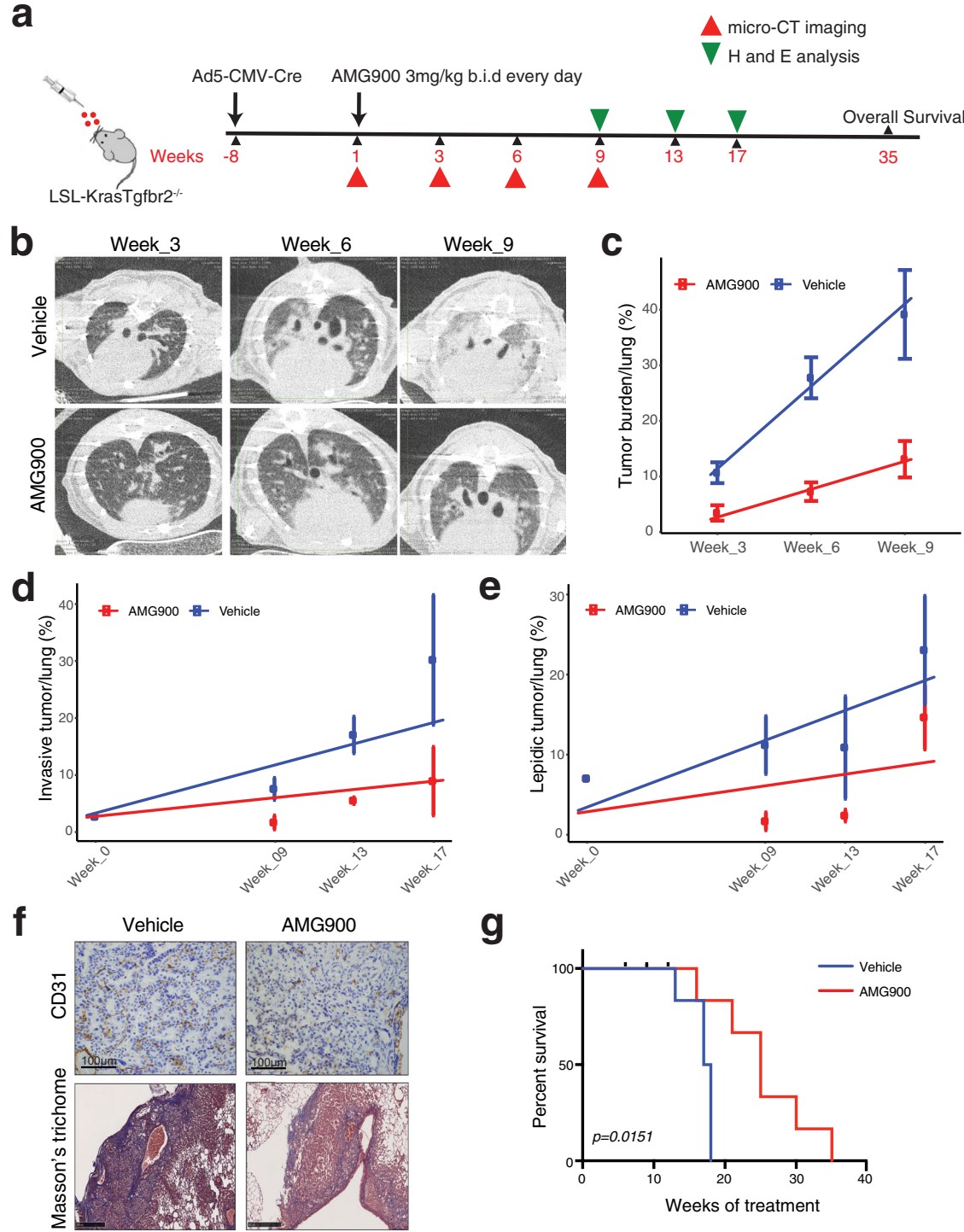

starting from week 5 after Ad5-CMV-Cre administration and initiated AMG900 treatment 8 weeks after Ad5-CMV-Cre instillation (Fig. 6a). AMG900 was well tolerated in the treatment group and there was no effect on animal weight (SFig. 11a) and there were no other signs of toxicity such as loss of fur, skin ulceration or decreased motility.

Micro-CT images demonstrate clear differences in tumor density and burden between vehicle and AMG900 treated animals at several time points of tumor progression (Fig. 6b). The tumor burden in vehicle group animals increases significantly faster compared to the tumor burden in AMG900 treated animals ($t$-value = −11.0 and $p = 1.4 \times 10^{-8}$ (Methods), Fig. 6c, SFig. 11b).

Histopathological analysis (weeks 9, 13, and 17) identified several invasive and in situ tumors with bulky nodules. While solid and acinar patterns increase in control animal lungs, lungs from AMG900 treated animals show suppression of invasiveness ($t$-value = −5.1 and $p = 4.0 \times 10^{-5}$ (Methods), Fig. 6d, SFig. 11c, d). The progression difference of lepidic tumor percentages between vehicle and treated animals is also significant ($t$-value = −3.0 and $p = 0.007$, Fig. 6e, SFig. 11c, d). The results together demonstrate the impact of aurora kinase on inhibiting tumor invasion. Because total tumor size is associated with proliferation, we performed an analysis of invasion extent in animals bearing similar tumor burden. Histopathological data

**Fig. 6 Aurora kinase inhibition suppresses invasive lung adenocarcinoma progression in Kras(G12D)/Tgfbr2$^{-/-}$ mouse model. a** Scheme of treatment of Kras(G12D)/Tgfbr2$^{-/-}$ mouse with AMG900. **b** micro-CT images of vehicle and AMG900 treated mice at week 3, 6, and 9 after treatment. Treatment was given at 3 mg/kg twice daily by oral gavage (vehicle $n = 10$, AMG900 $n = 8$). **c** Quantitation of tumor burden from micro-CT imaging for vehicle and AMG900 treated mice at week 3, 6, and 9 after treatment ($n = 3$). Center dots and error bars show means and standard deviation of tumor burden of each time point. A fitting line from a linear regression model (percentage ~ time) is shown for vehicle (blue) and AMG900 treated (red) mice. Source data are provided in Source Data files. **d** Comparison of invasive tumor percentage from vehicle and AMG900 treated mice between baseline and week 9, 13, and 17 from histopathological analysis ($n$ of vehicle = 2, 3, 5, 5; $n$ of treated = 2, 2, 3, and 3 for week 0, week 9, week 13, and week 17, respectively). Center dots and error bars show the means and standard deviation of tumor burden of each time point. A fitting line from a linear regression model (percentage ~ time) is shown for the vehicle (blue) and AMG900 treated (red) mice. Source data are provided in Source Data files. **e** Comparison of lepidic tumor percentage from vehicle and AMG900 treated mice between baseline and week 9, 13, and 17 from histopathological analysis ($n$ of vehicle = 2, 3, 5, 5; $n$ of treated = 2, 2, 3, and 3 for week 0, week 9, week 13, and week 17, respectively). Center dots and error bars show the means and standard deviation of tumor burden of each time point. A fit line from a linear regression model (percentage ~ time) is shown for the vehicle (blue) and AMG900 treated (red) mice. Source data are provided in Source Data files. **f** (Top) IHC images for CD31 staining (Scale bar 100 µm) in Vehicle and AMG900 treated mouse tumor. (Bottom) Masson's trichome staining (Scale bar 50 µm) in Vehicle and AMG900 treated mouse lung. **g** Survival curve for vehicle and AMG900 treated mice. (vehicle $n = 8$, AMG900 $n = 10$). LRT $p$-value was measured ($p = 0.015$). Source data are provided in Source Data files.

from animals in the vehicle and AMG900 treatment group shows a reduction in invasive tumor with aurora kinase inhibition, adjusted for tumor size, further supporting a direct link between aurora kinase inhibition and tumor invasion phenotype that is independent of tumor proliferation status (SFig. 11e).

In the human LUAD dataset, *TGFBR2* is included in the indolence signature and the indolence signature genes are significantly enriched for genes involved in the TGFB pathway (Supplementary Data 2), confirming the importance of TGFBR2 in tumor invasion and providing a potential model for assessing tumor cell aurora kinase signaling impact on the tumor microenvironment. In human cancer cells, RNAseq data show that aurora kinase inhibition suppresses the pro-invasive but not the indolence signature genes. When we compared differentially expressed genes between the mouse model and human data, genes upregulated in Tgfbr2 knockout mouse stroma are significantly enriched for the COL1A2 subnetwork (FET OR = 9.7 and $p = 6.1 \times 10^{-5}$) but not for the TPX2/AURKB subnetwork (FET OR = 1.2 and $p = 0.57$). In addition, AMG900 treatment in cancer cells suppresses most of the genes in the TPX2/AURKB subnetwork but not in the COL1A2 subnetwork. This indicates that expression changes of the TPX2/AURKB subnetwork likely reflect tumor intrinsic features while those of the COL1A2 subnetwork likely captures features of stromal cells in the tumor microenvironment impacted by alterations in tumor cell aurora kinase signaling.

We evaluated the impact of tumor cell aurora kinase inhibition on the tumor microenvironment composition. AMG900 treatment significantly reduced tumor-infiltrating CD31$^+$ neovessels as compared to vehicle-treated group ($t$-test $p = 0.03$) (Fig. 6f and SFig. 11f, g). Vehicle-treated tumors show heavy collagen deposition while AMG900 treated animals show significantly lower levels of collagen ($t$-test $p = 0.043$ Fig. 6f, SFig. 11h). We observed a loss in E-Cadherin expression in vehicle-treated invasive tumors, while AMG900 treated animals show uniformly strong expression of E-Cadherin ($p = 0.004$, SFig. 12a, c). Immunohistochemical staining for pAKT shows a strong positive expression in vehicle animal tumors, while the loss is observed in AMG900 treated animals ($p = 0.01$, SFig. 12d–f). AMG900 therapy led to a significant survival benefit over control group (LRT $p = 0.015$, Fig. 6g). While vehicle group animals died within 18 weeks of treatment, the AMG900 group survive for 35 weeks of treatment. Taken together, our in vivo transgenic data indicated that AMG900 treatment intercepts the progression of invasive lung adenocarcinoma and leads to better survival in this animal model of early-stage invasive lung adenocarcinoma.

## Discussion

In advanced lung carcinoma, genomic dissection of unique histological and clinical subtypes has identified actionable alterations that have led to the development and dissemination of targeted therapeutics that have remarkably altered the clinical course of this disease and have contributed to increased overall survival in lung cancer. Our premise is that similar approaches applied to early-stage lung cancer might be similarly impactful. Our initial step was to introduce and confirm the clinical and biological significance of lung adenocarcinoma invasiveness as a key driver of tumor progression, metastasis, and clinical outcome in early-stage disease. In this study, we generated signatures of lung adenocarcinoma invasion and indolence and validated the biological and clinical impact of these signatures in independent cohorts, human tissues, human cell lines, and in genetic mouse models of invasive lung adenocarcinoma. Using genomic networks and in-silico approaches to identify actionable therapeutic targets, we identified and validated inhibitors of aurora kinases A and B as effective interceptors of lung adenocarcinoma progression, invasion in cells and in the mouse model of invasive lung cancer.

Aurora kinases play important role in mitosis and are crucial in cellular processes such as chromosome segregation, chromosome alignment, and spindle assembly[50,56–58]. Our results in early lung cancer models show an important role for Aurora-A and Aurora-B in lung tumor progression through regulation of cell migration and invasiveness in lung adenocarcinoma. Aurora-A and Aurora-B function redundantly to influence lung cancer progression as shown by genetic deletion experiments showing that loss of Aurora-A and Aurora-B together, but not alone, leads to inhibition of migration and invasion of LUAD cells. Recent studies have shown Aurora-A and Aurora-B regulate cancer cell proliferation and tumor growth in other cancer models[44,46,59,60]. The role of aurora kinases in tumor migration and metastasis has been demonstrated in models of breast cancer[61–63], head and neck squamous cell carcinoma[64], glioblastoma[65], oral squamous cell carcinoma[66], and osteosarcoma[67,68]. Another report shows AURKB knockdown resulted in a reduction in migratory and invasive ability of osteosarcoma cell line by through mTOR/ULK1 pathway[69]. These studies support the generalizability of our findings to other cancer types and suggest a role for evaluating the impact of Aurora Kinase inhibition in the treatment of early extrathoracic malignancies. We show that the Akt/mTOR pathway mediates downstream signaling to influence tumor progression in Aur A/B inhibited cells and this suggests opportunities for combination strategies to address cells that may bypass aurora Kinase inhibition.

Aurora kinases may interact with driver mutations of tumor cells, however, in vitro assays showed that reduction of cell invasiveness by pan-aurora kinase inhibition is independent of KRAS mutation status. A recent study reported the role of driver mutation LKB1 and Aurora-A mediated phosphorylation of LKB1 in higher proliferation, invasion, and migration of NSCLC cells[70]. Among our panel of 8 highly invasive cell lines, only A549 harbors a LKB1 nonsense mutation and shows similar results with other LKB1 WT invasive cells from CRISPR mediated knockout as well as pan-inhibition of aurora kinase A and B, suggesting that the reduction in migration and invasion by deleting *AURKA* and *AURKB* is independent of LKB1 driver mutation status.

Using a genetically engineered mouse model, we previously reported *Tgfbr2* as a determinant of invasive phenotype of esLUAD, likely through modulating the tumor microenvironment that was enriched for expression of genes in the COL1A2 subnetwork and not for genes in the TPX2/AURKB signaling subnetwork[35]. It is important to note that tumor invasion phenotype in tumor tissue is complex depending on both tumor intrinsic molecular features and interactions between tumor cells and tumor microenvironment that may be mediated through aurora kinase signaling[71]. Further investigations including single-cell genomic studies will be necessary to elucidate how the tumor invasion phenotype is orchestrated by these subnetworks in a complex, interactive, and heterogeneous cellular environment.

Here, we show the potency of AMG900 in a genetically engineered mouse model of invasive LUAD with potential application to intercepting invasion in early-stage human lung adenocarcinoma. We mined genomic data acquired from human lung adenocarcinoma specimens to identify signatures of tumor invasiveness and we used network modeling approaches to identify key drivers susceptible to small-molecule inhibition in vitro and in vivo. Our preclinical studies exhibit abrogation of progression and spread of invasive LUAD tumor with AMG900 treatment thus illustrating the importance of aurora kinase inhibition as powerful therapeutic opportunity in clinic.

Clinical application of small molecule inhibitors is expanding for targeted therapy for several cancers. In lung adenocarcinoma, inhibitors to oncogenic drivers such as EGFR, BRAF and ALK have been in use to treat advanced lung adenocarcinoma for several years in clinic[72]. Recently, proof of principle for application to esLUAD was established in the ADAURA trial that demonstrated administration of adjuvant EGFR tyrosine kinase inhibitor (TKI) Osimertinib after surgical resection prevented recurrence and death[73]. We anticipate the initial clinical application for testing of the invasiveness signature and AMG900 will be in the setting of unresected subsolid lung nodules that are commonly detected as solitary or multiple lesions in patients screened for lung cancer by chest CT examination. The objective will be to determine lesions most at risk for progression and to determine if AMG900 interception reduces tumor progression and metastasis.

In summary, we report a gene signature capable of classifying esLUAD patients into indolent and invasive groups, and inferred aurora kinases as key regulators of the signature. We show that aurora kinase A and B together regulate migration and invasiveness in LUAD through activating Akt/mTOR pathway and EMT pathway. Our findings support stratification of esLUAD patients based on our gene signature and suggest a potential role of pan-aurora kinase small molecule inhibitors in esLUAD patients with tumors of an invasive gene signature.

## Methods

**Patient samples and histopathological analysis.** Tumor specimens were acquired from 53 patients (age: 66.9 ± 8.9 yrs., sex: female ($n = 33$, 63%)) of resected lung adenocarcinoma with histology classification that included AIS and MIA. The 53 frozen tumors tissues collected from 2000 to 2010 were retrieved for use of deidentified human tissue with clinical annotation as part of an Institutional Review Board (IRB) approved protocol of the Columbia University Cancer Center Tissue Bank (IRB #: AAAA-3987) that had waivers for consent. The human subjects protocol covered the use of banked tissue from deidentified individuals for genomic analysis and data dissemination and deposition. All cases of AIS, MIA, and LPA for which frozen tissue was collected were obtained. In addition, small invasive adenocarcinomas (prioritized as <2.0 cm, but all <3.0 cm) with node metastasis were identified (10 cases) and matched to 10 cases of the same T stage size grouping and with the same predominant invasive pattern but lacking node metastasis. Demographic details of these 53 patients are listed in Supplementary Table 1.

Tumor frozen sections were stained using 1% cresyl violet and were microscope-guided needle dissected in an RNase-free collection tip under vacuum suction to enrich for tumor >75% for further analysis. These samples were immediately processed for RNA extraction using the Qiagen RNeasy kit. RNA quality was assessed by Agilent Bioanalyzer, and tumors with RIN values <7.0 were excluded.

**RNA-sequencing for LUAD patients and cell lines.** Total RNA (~100–300 ng per sample) for each resected tumor sample was poly(A)-selected, fragmented, converted to cDNA, and barcoded for multiplex sequencing on the Illumina HiSeq 2500. The RNA sequencing reads were processed through a standard protocol for TopHat and Cufflinks[74]. For reference genome and transcriptome, hg19 reference genome and UCSC refseq gtf files were downloaded (hgdownload.soe.ucsc.edu/goldenPath/hg19/bigZips). The average number of reads per sample was 45 million and the mapped reads were further used to estimate the abundance of genes in Fragments per kilobase of transcript per million fragments (FPKM) (details in Supplementary Methods). For RNA sequencing of A549 and H1792 cells treated with DMSO or 0.1 μM AMG900 for 48 h, total RNA was purified from cells using Rneasy kit (Qiagen) for sequencing performed on NextSeq 500 (Illumina). Deseq2[75] was used to identify differentially expressed genes (DEGs) (FDR < 0.01) between control (DMSO) and AMG900 treated cells. Gene expression omnibus accession numbers: GSE166722 (GSE166720 for 53 primary tumors and GSE166721 for AMG900 treated cells).

**Identification of invasive signatures.** Unsupervised hierarchical clustering used the top 1000 most varying genes (variance > 1.0). Differentially expressed genes (DEGs) between the two clusters were determined by *t*-test (Fold-Change (FC) > 1.5 & FDR < 0.01). The samples were clustered again using the DEGs, and after two rounds of iteration, we reached the static state with no more change in DEGs or cluster memberships. The final clustering result contained two groups of 21 and 32 samples, labeled as "invasive" and "non-invasive", respectively, based on the predominant pathology histology in each group. The final signature genes were determined by *t*-test with the same cutoffs above and separated into "pro-invasive" and "indolence" signatures according to their direction across tumors.

For functional annotation, Hallmark gene sets, curated gene sets, and GO terms in Molecular Signatures Database (MSigDB) were used[21]. The significance of enrichment was determined based on FET *p*-value considering multiple testing (FET $p < 0.05/$(# of dataset tested)) in each category. For curated gene sets, "C2" collection in the MsigDB, we filtered gene sets using keywords such as "Lung cancer", "Epithelial mesenchymal transition (EMT)", "Invasion", "Migration", "Metastasis", and "Angiogenesis". Known tumor suppressor genes (TSGs) were collected from TSGene2.0[32].

**Gene expression data from lung tumors of KrasTgfbr2$^{-/-}$ mouse model.** Gene expression microarray data of 11 mouse lung tumors was downloaded from GSE27717. The dataset consists of five invasive lung tumors from KrasTgfbr2$^{-/-}$ mice and six non-invasive lung tumors from KrasTgfbr2$^{wt}$ mice[35]. The pro-invasive and indolence genes were used to cluster the murine tumors according to unsupervised hierarchical clustering. The DEGs between two groups of lung tumors were determined by *t*-test (FC > 1.5 and FDR < 0.01).

**Validation of the invasiveness signature in independent human lung tumor datasets.** We collected additional LUAD cohorts to test whether our invasiveness signature genes could classify samples into distinct subgroups associated with dissimilar patients' survival. For this purpose, we downloaded gene expression data from seven independent publicly available LUAD cohorts with survival information of the samples: a RNAseq data set from TCGA LUAD[76] and six microarray data sets including Shedden et al. (GSE68465)[22], Okayama et al. (GSE31210)[77], Der et al. (GSE50081)[78], Tang et al. (GSE42127)[79], Rousseaux et al. (GSE30219)[80], and Wilkerson et al. (GSE26939)[81]. We restricted our analysis to early-stage (I and II) tumor specimens. The number of samples used in this study is 371, 397, 204, 127, 111, 85, and 62 for Shedden et al., TCGA LUAD, Okayama et al., Der et al., Tang et al., Rousseaux et al., and Wilkerson et al., respectively. In addition to seven primary tumor datasets, we downloaded gene expression profiles of 70 LUAD cell lines from Cancer Cell Line Encyclopedia (CCLE) microarray data matrix[36].

**Estimation of invasiveness score (IVS).** First, using the pro-invasive and indolence signatures, we trained a classifier based on the elastic net to classify our original dataset whose invasiveness was determined (0 = indolent and 1 = invasive)[82]. To account for any potential differences between gene expression platforms, both the original and the testing gene expression matrix were z-transformed. The elastic net[83] is a regularized regression model that uses a linear combination of the $L_1$ and $L_2$ penalties of the lasso and ridge methods as below;

$$\min_{\beta_0,\beta} \left[ \sum_{i=1}^{N} \left( y_i - \beta_0 - x_i^T\beta \right)^2 + \lambda \sum_{j=1}^{|x_i|} \left( \frac{1-\alpha}{2}\beta_j^2 + \alpha|\beta_j| \right) \right] \quad (1)$$

where $y_i$ corresponds to binary class (invasive or non-invasive) and $x_i$ is a vector of features (z-transformed gene expression) for the $i^{th}$ samples. The $\beta$'s are regression coefficients that we estimate. The tuning parameter $\lambda$ is the weight of the regularization terms and is chosen to minimize mean square errors. $\alpha$ is the elastic net penalty. Using a R package, "glmnet", we performed cross validation to select the optimal regularized parameters of the elastic net as the elastic net penalty ($\alpha = 0.02$) and the regularization parameter ($\lambda = 0.1$). Then the elastic network classifier was applied to samples in each cohort to estimate relative IVS as predicted probability with distribution [0,1]. The association between IVS and patients' survival was tested using Cox regression with age, sex, and tumor stage as confounding factors.

$$\text{Survival} \sim \text{age} + \text{sex} + \text{stage} + \text{IVS} \quad (2)$$

For survival analysis in the TCGA LUAD dataset[76], we used overall survival (OS) data recently updated by Liu et al.[84]. For other datasets, vital status and the last contact or death days were used as deposited into Gene Expression Omnibus (GEO) database.

**Classification of samples based on IVS.** Samples in each independent primary tumor cohort were classified into three groups (invasive, intermediate, and indolent) based on IVS. We divided IVS into 40 bins (bin size = 0.025 between 0 and 1) and identified the smallest and largest local minima If IVS < smallest minima, the tumor was classified as "indolent", else if IVS > largest, the tumor was classified as "invasive". Otherwise, the tumor was classified as "intermediate" (SFig. 2a). The survival difference across these groups was compared. We censored the maximum time of survival at 5 years and measured the 5-year survival rates across tumor groups in each dataset with LRT $p$-value from a cox proportional hazards model. The "coxph" function from an R package "survival" was used to perform the survival analysis.

**Construction of a molecular causal network for esLUAD.** A molecular causal network for esLUAD was constructed by integrating methylation, CNV, and gene expression profiles of stage I patients in the TCGA LUAD dataset. RSEM data for gene expression, Illumina HumanMethylation450 matrix for DNA methylation, and CBS segment mean values for CNV were downloaded for stage I patients from the TCGA data portal (https://portal.gdc.cancer.gov). Sample alignment[85] was performed to confirm that different types of data were from the same individuals (details in Supplementary Methods) and 216 stage 1 tumor samples with all three types of data were included in the further analyses. A total of 8533 informative genes with detectable expression levels and large variances across samples were selected to be included in the network reconstruction process. Among them, the expression of 3476 and 761 genes were cis-regulated by CNVs or promoter methylation (FDR < 0.01), respectively, and cis-CNVs and cis-methylations were included as root nodes in the network construction as described in Supplementary Methods.

The gene expression, CNV, or methylation profile was discretized into three states: low, normal, and high level, guided by $k$-means clustering ($k = 3$) and biological meaningful cutoff values. The gene expression, cis-CNV, and cis-methylation nodes were then imported into the software suite, Reconstructing Integrative Molecular Bayesian Network (RIMBANet), to construct a biological causal network given the data and priors[37,38]. Briefly, the network reconstruction process searches for a directed acyclic graph (DAG) structure $G$ and associated parameters $\Theta$ that can best explain the given data $D$, $P(G,\Theta|D)$. If the structure $G$ is a DAG, then $P(G,\Theta|D)$ can be decomposed into a series of sub-structures.

$$P(G,\Theta|D) = \prod_i P(G^i,\Theta^i|D) \quad (3)$$

With cis-CNV and cis-methylation nodes included, the structures $X \rightarrow Y$ and $Y \rightarrow X$, given by;

$$p(X \rightarrow Y|D) = p\left(Y|X, \text{CNV}_y, \text{Methyl}_y D\right)p(X|D) \quad (4)$$

$$p(Y \rightarrow X|D) = p\left(X|Y, \text{CNV}_x, \text{Methyl}_x D\right)p(Y|D) \quad (5)$$

are no longer equivalent, so that potential causal relationships between $X$ and $Y$ can be inferred unambiguously. To speed up the searching process, for each gene, the bottom 20% of genes based on their mutual information were excluded as potential candidate regulators (sparse candidate search[86]). The network reconstruction process is a Monte Carlo Markov chain (MCMC) process. Given different random seeds, we might end up with different structures. Thus, we ran 1000 independent

MCMC processes based on 1000 random seed numbers that resulted in 1000 candidate structures. Then, we selected consensus structure features with posterior probabilities >0.3 among candidate structures[87]. Finally, loops in the consensus network were removed by deleting the weakest link in the loops. The resulting network was visualized using Cytoscape3.7[88]. Given a set of seed nodes $N_s$,

$$\text{SN}_s = \bigcup_i d(\text{node}, N_s^i) \leq l \quad (6)$$

is the union of nodes that are within $l$ steps from the seed node $N_s^i$, and the subnetwork for the seed nodes $N_s$ is the set of connections among $N_s$.

**Subnetworks enriched for invasive signatures.** Key driver analysis (KDA)[89,90] was performed to identify master regulators associated with pro-invasive or indolence signatures in two steps. First, for each node in the network, we extracted a list of nodes within two layers from the seeding node and tested the significance of enrichment against the pro-invasive or indolence signatures (FET $p$-value < $10^{-8}$, adjusting multiple testing). Sorted on FET $p$-values, a gene with the strongest $p$-value was determined as the top key regulator. Any candidate regulators in its two-layered neighbors were excluded from the candidate list. Then we identified the next key regulator with the strongest p-value among the remaining candidates. The process was iterated throughout the sorted candidate list.

To extract the pro-invasive and indolence subnetwork, we overlaid the pro-invasive and indolence signatures on the network and collected two large closed subnetworks; one with 123 genes, and the other with 97 genes. Drug treatment perturbation profiles curated in Connectivity Map (CMAP)[91,92] were used to investigate potential drug candidates that impact the expression of the genes in the two subnetworks. The CMAP database (LINCS database) encompasses 1.3 million L1000 profiles covering 8 different cell lines responding to 7977 drugs[43]. Using the 123 and 97 genes from the two invasive subnetworks defined above, we focused on results from the lung adenocarcinoma cell line A549 (https://clue.io/). The output small molecule classes were compared based on the negative enrichment score (enrichment score < −95).

**Human tissue microarray and immunohistochemistry.** TMAs of 768 consecutive lung adenocarcinoma cases from 1997–2000, 2002, 2010–2013 were constructed with 1.0 mm tumor plugs in triplicate with the exception of 2011, 2012, and 2013 which were in duplicate. The 2014 and 2015 arrays included Stage I adenocarcinoma only. We focused on 396 stage I tumors in our analysis. All TMA materials were accessed through the Columbia University Cancer Center Tissue Bank IRB (IRB #: AAAA-3987) and the Weill Cornell Thoracic Surgery Biobank IRB (IRB #: 1008011221) approved retrospective human tissue protocols and were constructed under those protocols that had waivers for consent. These specimens were not acquired from clinical trials and there is no provision for reidentification after the TMAs were constructed. The protocols covered analysis of archived specimens from deidentified individuals, data dissemination, and deposition. TMA patients have a similar demographic of age and sex with our esLUAD cohort (Supplementary Table 6). Immunohistochemistry for Aurora-A and Aurora-B in human TMA was performed using the Leica Bond system. Sections were pre-treated with heat-mediated antigen retrieval with Tris-EDTA buffer (pH = 9, epitope retrieval solution 2) for 20 min. They were incubated with primary antibodies for Aurora A or Aurora B for 30 mins at room temperature and detected using an HRP conjugated compact polymer system with DAB as the chromogen and hematoxylin as counterstain. Aurora A and B staining was measured by an H-score that integrates a semi-quantitative score of 0, 1 or 2+ for intensity and the percentage of positive cells. Nuclear staining for aurora kinase A was required in all cases, but when present, both nuclear and cytoplasmic staining was included for H-score. For aurora kinase B, only nuclear staining was scored. A previous study indicates certainty in nuclear localization for both proteins and suggestion of cytosolic staining for Aurora-A[93]. While there is data that Aurora-B has changing subcellular localization based on interaction with p53[48], cytosolic staining was not seen in our cases. Therefore, to best interpret both protein staining patterns, nuclear staining was required for both analyses. Since no cytosolic staining was seen for Aurora-B and was scored 0 in all cases it had no impact on H-score. The consistent presence of both nuclear and cytoplasmic staining for Aurora-A in these cases was seen and therefore both were included in the H-score.

We performed $t$-test to assess the significance of differences of Aurora-A and Aurora-B protein expression between non-invasive (AIS, MIA, and LPA) and invasive (AC, MP, SOL, and PAP) tumors. The association between the Aurora-A expression and survival was tested on patients with survival information (107 non-invasive and 158 invasive) using Cox regression with age, sex, and histology (invasive vs non-invasive) as confounding factors.

$$\text{Survival} \sim \text{age} + \text{sex} + \text{histology} + \text{AURKA} \quad (7)$$

**Cell culture.** Lung adenocarcinoma cell lines H1373 (CRL-5866), H1792 (CRL-5895), H2009 (CRL-5911), H1755 (CRL-5892), H1975 (CRL-5908), H1650 (CRL-5883), HCC-78 (ACC-563), H3255 (CRL-2882), Calu-3 (HTB-55), HCC-1833 (71833) and HCC2279 (CRL-2870) were maintained in RPMI 1640 (Gibco) supplemented with 10% FBS, 1% penicillin and 1% streptomycin. SK-LU-1 (HTB-57), A549 (CRM-CCL-185) and HEK293T (ACS-4500) cells were maintained in

DMEM (Gibco) supplemented with 10% FBS, 1% penicillin and 1% streptomycin at 37 °C and 5% $CO_2$. All cell lines were regularly tested for mycoplasma using the mycoAlert Detection Kit (Lonza). HCC-78 was obtained from DSMZ—German Collection of Microorganisms and Cell Cultures GmbH, Braunschweig, Germany. HCC-1833 was obtained from Korean cell line bank, Seoul, Korea. All remaining cell lines were obtained from American type culture collection (Manassas, VA).

**Lentivirus production.** HEK293T cells were seeded in 10 cm culture dish. After reaching confluency of 70–80%, cells were co-transfected with 10 μg target plasmid construct, 7.5 μg of psPAX2 (Addgene #12260) and 2.5 μg pMD2.G (Addgene #12259) vectors using TransIT®-Lenti Transfection Reagent (Mirus) according to protocol. Lentivirus particles were harvested after 48 h of transfection and filtered with 0.45 μM filter for transduction into target cells.

**CRISPR-Cas9 genome editing.** Lenti-Cas9-Blast plasmid (Addgene #52962) or hUBCp_Cas9_3xNLS_p2a_puroR plasmid (Addgene #81251) were used to generate cells with stable human S. pyogenes Cas9 expression. Cells were infected with lentivirus and supplemented with polybrene (Sigma) at a final concentration of 8 μg/ml. Transfected cells were selected with 8–10 μg/ml of blasticidin or 3–5 μg/ml puromycin for 6–10 days. For generating CRISPR knock out cell lines, short guide RNA (gRNA) target sequences for *AURKA* and *AURKB* and non-target sgRNAs were acquired from human CRISPR knockout pooled Brunello library[94]. sgRNAs were cloned into pLKO.1 GFP or pLKO.1 mCherry vector at Bbs1 site downstream of the human U6 promoter. pLKO.1-GFP plasmid was a gift from Dr. Brian D Brown laboratory (ISMMS, New York, NY) and pLKO.1 mCherry was purchased from Addgene (#128073) (Supplementary Table 9 for plasmid details). Sequences of sgRNAs are listed in Supplementary Table 10. Cas9 expressing cells were infected with pLKO.1-GFP-sgRNA or pLKO.1-mCherry-sgRNA.

**Migration and invasion.** For migration assay, $50 \times 10^3$ cells were seeded on an 8 μM cell culture insert (Fisher Scientific) in triplicate wells in serum-free media in a 24-well plate and 800 μL of 10% FBS supplemented media was added to the lower compartment of the well. Cells were incubated for 48 h in a 37 °C incubator and then the inserts were washed with phosphate-buffered saline. Cells on the top of the transwell were scraped and washed away using a cotton tip applicator. Cells on the bottom side of the transwell were fixed with 70% ethanol and stained with 0.2% crystal violet. Images were analyzed using ImageJ software for each replicate.

For invasion assay, 8 μM cell culture insert was coated with 300 μg/ml of Corning matrigel basement membrane matrix (Corning) and incubated for 1 h in 37 °C incubators. $50 \times 10^3$ cells were then seeded over the layer of matrigel in the 8 μM cell culture insert and was stained in a similar fashion as migration assay. To test whether phenotype changes were significant or not, we measured the significance of a fit line from a generalized linear regression model ($n = 3$);

$$percentage(invasion \ or \ migration) \sim dose \quad (8)$$

All tests were significant (two-side test $p < 0.05$) and results are reported in Supplementary Table 7.

**Wound healing assay.** Cells were cultured to 100% confluency in a 24-well plate and the wound was carefully created using P20 pipette tip. Images of the wound were captured at 0, 24, 48, and 72 h to evaluate the closure of the wound. Images were analyzed using ImageJ software. A linear model

$$percentage(wound \ healing) \sim dose \quad (9)$$

was used to test the significance of phenotype changes in a dose-dependent manner. Results are reported in Supplementary Table 7.

**In vivo transgenic mouse model and drug treatment.** All mouse studies were conducted in compliance with regulations and guidelines from Institutional Animal Care and Use Committee (IACUC# LA11-00201) from the Icahn school of Medicine at Mount Sinai (ISMMS), NY. Mice were maintained under pathogen-free conditions at ISMMS, NY with ambient temperature of 22–25 °C and relative humidity of 50%. All mice were housed under a 12 h light/12 h dark cycle as per institutional IACUC regulations and the health of mice was monitored daily. For animal transgenic studies, C57/Bl6 male or female mice at age 6-8 weeks were used. We generated transgenic mouse LSL-KrasTgfbr2$^{-/-}$ by crossing LSL-KrasG12D positive mice with Tgfbr2$^{flox/flox}$ mice[35]. Mouse genotypes were confirmed at 21 day from birth by PCR amplification of genomic DNA isolated from tail snips (primer sequence available on request). To induce tumor formation, $2.5 \times 10^7$ pfu particles of Ad5-CMV-Cre (University of Iowa) were administered intranasally to LSL-KrasTgfbr2$^{-/-}$ mice at the age of 6-8 weeks[35]. Adeno-Cre viral particles were suspended in serum-free DMEM and were mixed with 2 M $CaCl_2$ 20 minutes before administration. To begin AMG900 treatment, KrasTgfbr2$^{-/-}$ mice were imaged by micro-CT every week starting from 5 weeks to identify the earliest signs of tumor formation. At week 8 post Ade Cre administration animals were randomly divided into two groups and treated with either vehicle (0.5% hydroxy propyl methyl cellulose, 0.1%

Tween 80, pH = 2.2) or 3 mg/kg AMG900[95] by oral gavage twice every day ($n = 10$ per group). Mice were sacrificed at several time points during treatment for histopathological analysis for tumor lesion comparison. The tumor burdens of mice from two groups (vehicle and AMG900 treated) were compared using a generalized linear model,

$$percentage(tumor \ burden) \sim time + time : group \quad (10)$$

and significance of the interacting term (time:group) was measured with $t$-value and $p$-value. Since tumor size cannot be externally monitored in a transgenic mouse model, animals were monitored for any signs of distress due to tumor burden such as loss of weight exceeding 20% of normal, body condition score ≤ 2, animals no longer able to access food or water or animals that are unresponsive, according to institutional IACUC regulations.

**Micro CT imaging and quantitation.** KrasTgfbr2$^{-/-}$ mice were imaged regularly starting from week 5 to monitor tumor spread by micro-CT imaging. For micro-CT imaging, animals were anesthetized using 100 mg/kg ketamine and 10 mg/kg xylazine injected via intra-peritoneal injection at the dose of 0.1 g per 10 g of body weight. Mice were intubated with 20 G × 1 inch catheter with the help of Fiber optic lightening kit (Kent Scientific Corporation, CT). Mice were ventilated on a small animal ventilator MiniVent (Hugo Sachs Electronik, Germany) and imaged on nanoScan PET/CT (Mediso USA). Images were acquired with a 300 ms scan at 150 kVp and 610 μA. 3 to 5 mice were imaged per group at several time points before and after starting AMG900 treatment. Horos v3.3.6 was used to mark the tumor region of interest and total lung and % tumor burden was calculated for each lung.

**Histopathological analysis.** Lung sections from LSL-KrasTgfbr2$^{-/-}$ animals treated with vehicle or AMG900 collected at several time points were formalin fixed paraffin embedded and stained with hematoxylin and eosin for analysis by expert pathologist. Each lung section was carefully studied by pathologist and marked for invasive and lepidic tumor areas. Lepidic and invasive tumor area per lung was calculated using Aperio ImageScope 12.1 software.

**Reporting summary.** Further information on research design is available in the Nature Research Reporting Summary linked to this article.

## Data availability

RNA sequencing data generated in this study is deposited in GEO database and raw files of the 53 human primary tumors and cell lines experiments are available with the accession code GSE166722. Publicly available datasets used in this study are GSE27717 (Kras/Tgfbr2 mouse model[35]), GSE68465 (Shedden et al.[22]), GSE31210 (Okayama et al.[77]), GSE50081 (Der et al.[78]), GSE42127 (Tang et al.[79]), GSE30219 (Rousseaux et al.[80]), and GSE26939 (Wilkerson et al.[81]). RNAseq, CNV, and DNA methylation profiles of TCGA Stage I and II LUAD samples were downloaded by GDC data portal [https://portal.gdc.cancer.gov/projects/TCGA-LUAD]. Microarray genes expression profiles of 70 LUAD cell lines were downloaded from CCLE Data section [https://depmap.org/portal/download/?release=CCLE+2019&release=Fusion&release=DNA+Copy+Number]. Source data are provided with this paper. The remaining data are available within the Article, Supplementary Information or Source Data file. Source data are provided with this paper.

## Code availability

All codes and scripts used in this study are available in Dr. Zhu's lab git repository https://github.com/integrativenetworkbiology/Tumor_invasion_esLUAD.

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

## Acknowledgements

C.A.P. is supported by NIH (R01CA163772 and R01HL130826) and NYSTEM C34052GG. H.W. is supported by the ATS Foundation Unrestricted Grant (ATS-2017-24), the American Lung Association of the Northeast Lung Cancer Discovery Award (LCD-504985), Department of Defense (W81XWH-19-1-0613), and NIH (R01CA240342). We acknowledge experimental contributions by Youngjae Woo, Yi Zhong, Avinash Kumar, and Anait S. Levenson. We thank the BioMedical Engineering and Imaging Institute and the Pathology Core Facility at ISMMS.

## Author contributions

S.Y., A.S., A.C.B., J.Z., and C.A.P. designed the experiments. S.Y., A.S., D.Y., N.K.A., R.T., W.W., D.C., E.L., A.S.P., T.S., R.K., B.D., and E.E.S. performed the experiments. S.Y., A.S., H.W., A.C.B., and J.Z. analyzed the data. A.C.B., P.P.M., and C.A.P. contributed to patient sample collection. S.Y., A.S., J.Z., and C.A.P. wrote the paper. All authors edited and critically reviewed the paper and agree to the final version of the manuscript.

## Competing interests

S.Y., E.L., E.E.S., and J.Z. are employees of Sema4, a for-profit organization that promotes personalized patient care through information-driven insights. C.A.P. reports consulting fees from Astra Zeneca, BMS, Daiichi Sankyo, and Ethicon outside the submitted work. Other authors declare that they have no competing interests.
