## [Peer review file · Nature Communications]

REVIEWER COMMENTS

Reviewer #1 (Remarks to the Author): Expert in lung cancer Aurora kinase therapy, interactomes, and genomics

Summary: In this manuscript by Yoo et al., the authors utilize RNA sequencing and network analysis techniques to gain insight into the molecular mechanisms of early-stage lung adenocarcinoma (esLUAD) invasiveness. This is an important question for clinical settings that were recently approached by Wang, S. et al. (Nature Communication, 2020)¹. Authors of the manuscript in particular are focused on differentially expressed genes identified using 53 early-stage LUAD patient samples. Through this data, the authors aim to detect genes associated with invasiveness of the LUAD. Identified gene signature is associated with low survival probability across multiple cohorts. Network analysis suggested Aurora A and Aurora B as regulatory proteins in this signature. Finding a predictive signature and studying mechanisms of invasiveness in early-stage LUAD can facilitate the intervention of invasive LUAD development. However, we have multiple concerns regarding the manuscript, including the interpretation of experimental data.

Overall, we think the analysis itself and the experimental design addresses an essential aspects of early-stage LUAD, but fundamental concerns in data interpretation jeopardize the interpretation of the key results. The central concern that we have with the conclusion of the paper is that it remains unclear if Aurora Kinases mediate invasion in a manner independent of its well documented regulation of proliferation. This is important because proliferation is a well documented predictor of survival.

Major concerns:

1. The authors claim that the signature defined by gene expression driven clustering is informative in identification of invasive phenotypes. What is the advantage of this approach over using histology alone? There is no direct comparison with the set of genes identified by the histology-based approach. Strong overlap of signature defined by gene expression driven clustering with histology-based and N.stage-based signatures (Fig. 1c) indicates that clustering-based signature might be not specific to invasiveness per se but could be a complex tumor characteristics for overall aggressiveness including tumor growth and/or metastatic potential as well as invasiveness. What is the evidence that this signature is specific for invasion rather than other tumor phenotypes?

2. Earlier work shows that most published gene prognostic signatures, not matter how fancy and tailored, reflect tumor proliferation rather than specific biology (<https://doi.org/10.1371/journal.pcbi.1002240>). Can you show this signature is independent of tumor proliferation? The genes selected in the invasion signature appear reflective of tumor proliferation (e.g. G2M, E2F).

3. Figure 2 shows samples in each independent primary tumor cohort that are classified into three groups (invasive, intermediate, and indolent). Classification based on IVS that is identified through the smallest and largest local minima is unusual and points to possible lack of robustness in the results. Supplementary Figure 2A indicates that histogram are quite noisy due to probably small sample size in several data sets. This can rise the concern of noise-driven cutoff selection and classification. Floating cut-off could bring bias in classification and influence survival analysis interpretation. The authors should perform such analysis with either fixed or standard percentile based cutoffs to demonstrate their findings are robust.

4. Although 70 cell lines were used for characterization of IVS, selected 5 cell lines from highly invasive group (NCI-H1373, SK-LU-1, NCI-H1792, A549, and NCI-H2009) are harbor mutant K-Ras. Whereas less invasive cell lines (HCC78, NCI-H3255, Calu3, HCC1833, HCC2279) are characterized by p53 and are KRAS wild-type. What is the selection criteria for cell lines to confirm invasiveness? Do highly scored cell lines without K-ras mutation (such as H1755, b-Raf mutant are associated with increased migration and invasion?

5. A possible confounder in the analysis Fig 2b are the growth rate of the selected cell lines, and the focus on cell cycle kinases in the rest of the manuscript suggests this is a possibility. Is there difference in proliferation rate between Kras mutant cells (NCI-H1373, SK-LU-1, NCI-H1792, A549, and NCI-H2009) identified as highly invasive and K-Ras WT cells (HCC78, NCI-H3255, Calu3, HCC1833, HCC2279) classified as less invasive? This is important to demonstrate that the results are independent of other aggressive tumor phenotypes, and specific to invasion.

6. The authors focus on the role of AURKA and AURKB in lung tumor progression and speculate that they regulate cell migration and invasiveness in lung adenocarcinoma. The major concern is that the observed effect on cell migration and invasion assays might be due to reduction in cell proliferation and survival upon manipulation of AURKA and AURKB expression. Is the proliferation rate of cells effected by AURKA and AURKB downregulation? Nearly every published experiment of genetic knockout of either has significant effects on cell growth (e.g. see depmap database). The proliferation rate should be measured for all cell lines generated to estimate the effect of gene knockdown on cell proliferation and to clearly demonstrate migration/invasion effects are independent of proliferation.

7. It is known that KRAS positively regulates AURKA and AURKB expression in LUAD cell lines and Aurora kinase inhibition was suggested as a novel approach for KRAS-induced lung cancer therapy in 2016². More importantly, dual pharmacological inhibition of AURKA and AURKB reduced growth, viability, transformation, and induced apoptosis in vitro in an oncogenic KRAS-dependent manner including A549 cell line, that was also included in current study. Throughout the whole study only K-ras mutant cell lines were utilized to study the expression of AURKA and AURKB and the effect of their manipulation on different physiological outcomes including cell migration and invasion. By experimental design the invasive signature is defined regardless of mutation status. Moreover, Figure 2 shows that cell lines that are known by their wild type K-ras status are also assigned with high invasive score (for example H1755,

PC19, H1975). The effect of AURKA and AURKB manipulation should be tested in KRAS wild type cell lines.

8. Figure 6 and extended data figure 10 show clear differences in tumor burden and change between vehicle and AMG900 treated animals at several time points of tumor progression. This might be also explained by effect in tumor cell proliferation because at week 3 there is no visually detectable tumor sites. To neglect the effect of proliferation/survival on the tumor development in vivo (figure 6) and to clearly link the effect of AMG900 inhibitor with effect on invasiveness it will be informative to compare percentage of invasive tumors across mice that harbor matching tumor burden. Overall the animal data are not compelling and the differences in Fig 6d,e do not appear significant.

Minor Concerns:

1) Data visualization requires improvement. To simplify interpretation, and to visualize patterns in the complex network, networks might be grouped in according to their link to inhibitor term or GO term. Extended data figure 4 does not show any useful information and requires significant improvement.

2) P values might be displayed in the figures, and statistical analysis at least mentioned in the figure legends. Figure 4E should contain statistical data. For example, cells containing statistically significant change can be labeled with (*)

1 Wang, S. et al. Tumor evolutionary trajectories during the acquisition of invasiveness in early stage lung adenocarcinoma. *Nature communications* 11, 6083, doi:10.1038/s41467-020-19855-x (2020).

2 Dos Santos, E. O., Carneiro-Lobo, T. C., Aoki, M. N., Levantini, E. & Basseres, D. S. Aurora kinase targeting in lung cancer reduces KRAS-induced transformation. *Mol Cancer* 15, 12, doi:10.1186/s12943-016-0494-6 (2016).

Reviewer #2 (Remarks to the Author): Expert in lung cancer therapy and mouse models

Title: Integrative Network Analysis of Early-Stage Lung Adenocarcinoma Identifies Aurora Kinase Inhibition as Interceptor of Invasive and Progression

Summary: The authors have created a computational pipeline analyzing their in-house RNA-seq of early-stage lung adenocarcinoma (n = 53) to determine the signatures associated with invasive and indolent groups. The signature gene sets were validated by the expression data of the *Kras*⁺/*Tgfbr2*^{-/-} genetically engineering mouse model. They established the Invasiveness score (IVS) based on these signature genes to predict prognosis of stage I lung adenocarcinoma (LUAD) patients in 7 published available cohorts. Aurora kinases (AURKA and AURKB) were identified as regulators of the pro-invasive network and validated as highly expressed by tissue microarray. The authors demonstrated that Aurora kinase inhibitor, AMG900, reduced the migration and invasiveness of human cell lines with high IVS and the invasive mouse model.

The authors analyzed multiple data sets, including human, mouse model and cell lines, to support their findings. This is an important area of investigation of potential clinical relevance. However, the finding of Aurora kinases as key regulators has been demonstrated in multiple lung cancer studies (summarized in PMID: 33202573).

Major comments

1. In line 125-126, the authors claim that their signature genes based on clustering approach are “more informative” than comparisons based on histologically defined groups. The comparison between two histology groups defines 793 DEGs, while the clustering approach provides 1322 genes, representing a 40% increase in genes. The higher number of candidates often improves the prediction but may also increase the chance of false positives and overfitting. To demonstrate that the clustering approach provides a gene set with more biological meaning than phenotype-comparisons, authors should build the IVS based on the same number of genes for each approach and then compare the predictions.
2. The authors utilized *Kras*⁺/*Tgfbr2*^{-/-} genetically engineering mice to validate pro-invasive signature genes identified in human data and assess the expression changes induced by the aurora kinase inhibitor. However, in the original study of the mouse model, Borczuk and colleagues (Ref 32) claim that *Tgfbr2* deletion changed the tumor microenvironment and an induced invasive phenotype. Therefore, the authors should discuss if the human data showed deregulation of the TGFBR pathway or were enriched by DEGs associated with changes in the tumor environment based on the mouse model.
3. It has been shown that KRAS positively regulates AURKA expression (PMID: 26842935). Both cell lines A549 and H1792 utilized to evaluate aurora kinase inhibitors have KRAS mutation. The authors should address the relationship between AURKA/B and driver gene mutations. If the high expression of AURA/B in the invasive group is independent of mutations in RAS pathways, they should evaluate the therapy in other human cell lines without KRAS mutations. A549 also has an inactivating LKB1 co-mutation. Given

the previously published relationship with Aurora Kinase and LKB1 (PMID: 28967900), the authors should include discussion of its potential role.

4. The authors claim the AMG900, aurora kinase inhibitor, impairs AKT/mTOR and EMT pathways. However, they did not discuss or demonstrate if the impact was via aurora-kinase dependent or independent pathways. Were the AKT/mTOR and EMT markers also changed in A549 and H1792 with CRISPR/Cas9 deletion AURKA/AURKB?

5. In figure 3e, the survival curve of patients with highest AURKA score (score =2) was not distinguished from the lowest score. How are the KM curves based on two groups (0 vs. > 0)? What is the outcome based on AURKB?

6. Aurora kinases are known to be associated with cell proliferation. The growth rate or the doubling time of a cell line will depend on its aurora kinase level. The migration and invasive assay are at least 48 hours in duration, which is longer than the doubling time of the cell lines utilized (e.g., 22-24hrs for A549). Therefore, the authors should shorten the timeframe (t = 24hr) for these assays to eliminate the effect of cell proliferation following migration through the membrane.

Minor comments

7. The methods (line 616-619) indicate that AURKA and AURKB were scored differently based on their nuclear or cytoplasmic location in the TMA data analysis. The authors should explain the biological meaning for the deviation of the scoring system.

Reviewer #3 (Remarks to the Author): Expert in lung cancer clinical research, lung cancer invasion and genomics

The paper from Seungyeul Yoo et al identifies aurora kinase as one of key regulators of invasiveness in early stage lung adenocarcinoma and inhibition of aurora kinases attenuate tumor invasion both in vitro and in vivo. Collectively, authors' key findings include: a) generate gene signatures which distinguish invasive and indolent lung adenocarcinoma and validated the functions of these gene signatures in independent cohorts, human tissues, cell lines and genetic mouse models. b) identifies aurora kinase inhibition as therapeutic approaches to block cancer invasion and progression in cells and animal models. The authors make a great effort to distinguish patients with high invasive properties even at early stages and provides potential therapeutic targets for treatment, which fills the gap and be of great interest to the clinicians in this field. However, several terms are not very convincing which need to be addressed in the revised version.

1. Line 106, about the definition of "invasive" and "indolent". Here, two distinct groups were identified based the clusters. Cluster in group with pathologically aggressive subtype was defined as "invasive", cluster in group with pathologically less invasive subtypes was defined as "indolent". If this definition was finally based on pathological classification, then why not do differential cluster analysis according to

pathological classification first. Again, in line 268, categorizing tumors into indolent and invasive groups was based on their histology which is not consistent with first one. Different classification of "invasive" and "indolent" were applied in this paper. It need to clarified.

2.Line 93, 53 early stage lung adenocarcinoma samples were analyzed. How about driver gene profile for those patients? Cancer cells with different driver gene profile have quite distinct biological characteristics which might contribute to the invasiveness.

3.Line 172. In this paragraph, gene signature was further validated in 7 databases by overall survival analysis. The association of gene signature with survival should also be analyzed in his own patient group.

4.Line 71, type 2 TGF Beta Receptor was identified as determinant of invasiveness and metastasis of localized lung adenocarcinoma. In current study, TPX2 and AURKB were found as key regulator of pro-invasiveness. Then, how to explain different findings?

5.Line 327, DEGs were highly consistent in two cell lines treated with aurora inhibitors. Then how about consistence of these DEGs with gene cluster identified in figure 1?

6.Line 337 and line 388, In cell lines, no significant changes of COL1A2 subnetwork were found, while in animal models, obvious changes of collagen were found after treatment of AMG900. How to explain this? At least, it should be explained in the discussion part.

7.Line 387, inhibition of Aurora with AMG900 were associated with neo-angiogenesis. The gene signatures were associated with cell cycle, EMT and angiogenesis when compared with Hallmark gene sets (paragraph 2, line 142). Thus, it is better to have more analysis including cell cycle and EMT changes after treatment of AMG900 in animal models?

8.Line 415, small spelling mistake: squamous cell carcinoma.

REVIEWER COMMENTS

Reviewer #1 (Remarks to the Author): Expert in lung cancer Aurora kinase therapy, interactomes, and genomics

Summary: In this manuscript by Yoo et al., the authors utilize RNA sequencing and network analysis techniques to gain insight into the molecular mechanisms of early-stage lung adenocarcinoma (esLUAD) invasiveness. This is an important question for clinical settings that were recently approached by Wang, S. et al. (Nature Communication, 2020)¹. Authors of the manuscript in particular are focused on differentially expressed genes identified using 53 early-stage LUAD patient samples. Through this data, the authors aim to detect genes associated with invasiveness of the LUAD. Identified gene signature is associated with low survival probability across multiple cohorts. Network analysis suggested Aurora A and Aurora B as regulatory proteins in this signature. Finding a predictive signature and studying mechanisms of invasiveness in early-stage LUAD can facilitate the intervention of invasive LUAD development. However, we have multiple concerns regarding the manuscript, including the interpretation of experimental data.

Overall, we think the analysis itself and the experimental design addresses essential aspects of early-stage LUAD, but fundamental concerns in data interpretation jeopardize the interpretation of the key results. The central concern that we have with the conclusion of the paper is that it remains unclear if Aurora Kinases mediate invasion in a manner independent of its well documented regulation of proliferation. This is important because proliferation is a well documented predictor of survival.

Major concerns:

1. The authors claim that the signature defined by gene expression driven clustering is informative in identification of invasive phenotypes. What is the advantage of this approach over using histology alone. There is no direct comparison with the set of genes identified by the histology-based approach. Strong overlap of signature defined by gene expression driven clustering with histology-based and N.stage-based signatures (Fig. 1c) indicates that clustering-based signature might be not specific to invasiveness per se but could be a complex tumor characteristics for overall aggressiveness including tumor growth and/or metastatic potential as well as invasiveness. What is the evidence that this signature is specific for invasion rather than other tumor phenotypes?

We appreciate the reviewer's comment. To demonstrate the advantage of gene expression driven signatures in these tumors, we directly compared these signatures to those driven by histology alone.

First, we compared the statistical significance of the expression changes of the 727 overlapping genes included in both gene expression-driven and histology-based signatures (Figure 1c). Although the genes identified by both approaches were significantly differentially expressed between invasive and non-invasive tumors (FDR<0.01), the t-test FDRs (the Benjamini-Hochberg adjusted p-values) of the genes from the gene expression-based clustering were more significant compared to those from the histology-based grouping for both upregulated and downregulated genes (**Reviewer Figure 1**), indicating that expression-driven clustering provides more robust tumor groupings than histology alone. The figure is added as **Supplementary 1d**.

Reviewer Figure 1. Gene expression differences of 727 common signature genes identified by both gene expression-driven clustering and histology-based classification. Left: 288 common signature genes in the pro-invasive signature and upregulated genes in histologically invasive tumors. Right: 439 common signature genes in indolence signature genes and upregulated genes in histologically non-invasive tumors. T-test FDRs of these genes between invasive and non-invasive tumors were compared between two approaches of tumor classifications.

Second, we compared the significance of overlaps between the MSigDB gene sets (Figure 1d) with the histology-based DEGs vs. the expression-driven invasion signature genes. Both signatures were significantly enriched for multiple MSigDB gene sets. As measured by Fisher's Exact test (FET) p-values, more significant overlap/enrichment was observed with the invasion signature genes (**Reviewer Figure 2**). Moreover, while genes identified by the two approaches were similarly enriched for cell cycle related gene sets such as G2M_CHECKPOINT (FET $p = 2.8 \times 10^{-52}$ and 2.2×10^{-51} for gene expression-driven and histology-based signatures, respectively) and E2F_TARGETS (FET $p = 1.1 \times 10^{-54}$ and 7.8×10^{-50} respectively), the significance of the enrichments for tumor invasion related gene sets was different. For example, HALLMARK_EMT (FET $p = 9.9 \times 10^{-29}$ and 5.8×10^{-14} for pro-invasive signature genes and histology-based signatures, respectively), SCHUETZ_BREAST_CANCER_DUCTAL_INVASIVE_UP (FET $p = 1.5 \times 10^{-25}$ and 2.6×10^{-13} , respectively), and ANASTASSIOU_MULTICANCER_INVASIVE_SIGNATURE (FET $p = 18 \times 10^{-37}$ and 5.8×10^{-19} , respectively). Similarly, the indolence signature by gene expression-driven clustering showed more significant enrichment for POOLA_INVASIVE_BREAST_CANCER_DN (FET $p = 3.1 \times 10^{-6}$ and 0.007 for gene expression-driven and histology-based signatures, respectively) and Tumor suppressor genes (FET $p = 3.5 \times 10^{-8}$ and 7.1×10^{-5} , respectively). The figure is added as **Supplementary Figure 1f**.

Reviewer Figure 2. Functional enrichment analysis against MSigDB gene sets for signatures identified by expression-driven clustering and histology-based classification. Left: Genes upregulated in invasive tumors. Right: Genes upregulated in non-invasive tumors. Fisher's exact test FDRs were used to assess the significance of enrichment against the gene sets.

Third, genes uniquely identified in the pro-invasive signature but not in the histology-based signature (236 genes in Figure 1c) were significantly enriched for gene sets related to tumor invasion and metastasis (**Reviewer Figure 3**), suggesting that signature genes uniquely identified by expression-driven clustering are specifically enriched for tumor invasion and metastasis ontology. This figure is added as **Supplementary Figure 1g**.

Reviewer Figure 3. Gene sets significantly enriched in the signature genes uniquely identified in the pro-invasive signatures but not in histology-based DEGs.

Taken together, these results suggest that the expression-driven clustering approach identifies signature genes with more functionally significant association with tumor invasion. Detailed results described above are included in Supplementary Information on page 6. In the revised manuscript, we added the following on page 6 and 8:

Page 6: “Compared to histology-based signatures, the expression differences of the overlapping genes between the signatures by the two approaches were more significant in the expression-driven approach (SFig. 1d) suggesting that tumor classification by gene expression-driven clustering is more robust for comparing invasiveness phenotypes among esLUAD tumors than tumor classification by histology or node metastasis status alone.”

Page 8: “We also compared functional enrichment of the expression-driven clustering signatures with those of histology-based DEGs (Supplementary Information). Although both gene signatures were enriched for multiple common MsigDB gene sets including cell cycle related gene sets, more significant enrichment for tumor invasion related gene sets was generally observed with the pro-invasive signature genes (SFig. 1f). Moreover, genes uniquely identified in the pro-invasive signature were significantly enriched for tumor invasion related gene sets such as MULTICANCER_INVASIVE_SIGNATURE (FET OR= 27.25 and $p= 9.9 \times 10^{-17}$) and EMT (FET OR= 10.9 and $p= 1.4 \times 10^{-17}$) (SFig. 1g) suggesting that the expression-driven clustering approach not only increases the number of genes of interest but also captures unique

molecular features related to tumor invasiveness compared to histology- or node metastasis status-based classification.”

2. Earlier work shows that most published gene prognostic signatures, no matter how fancy and tailored, reflect tumor proliferation rather than specific biology (<https://doi.org/10.1371/journal.pcbi.1002240>). Can you show this signature is independent of tumor proliferation? The genes selected in the invasion signature appear reflective of tumor proliferation (e.g. G2M, E2F).

We agree that cell proliferation contributes to survival/prognosis and invasion-related phenotypes, as shown above in Reviewer Figure 2. Indeed, our pro-invasive gene signature included 57 out of 131 genes in the tumor proliferation meta-PCNA signature (OR=36.8, FET $p=5.3\times 10^{-57}$) from Venet et al, which the reviewer referenced. When we used the meta-PCNA genes to cluster the 53 primary tumors, some of invasive tumors were clustered together with non-invasive tumors and *vice versa* (Reviewer Figure 4). Moreover, when we cluster the tumors using our signature *excluding* the meta-PCNA signature genes, the clustering result was completely agreed with the original clustering shown in Figure 1a (Reviewer Figure 4), suggesting that tumor proliferation alone is not sufficient to explain the molecular differences between invasive and non-invasive tumors and our signature genes capture tumor invasive specific biology in addition to tumor proliferation. This figure is added as Supplementary Figure 1e.

Reviewer Figure 4. Tumor clusters determined by meta-PCNA genes (left) and our signature genes excluding meta-PCNA genes.

In the revised manuscript, we added the following on pages 7 - 8:

Pages 7-8: “Tumor proliferation is a fundamental component of tumor malignancy and a well-known phenotype contributing to patients’ prognosis, thus it is expected that cell cycle genes would be represented in our invasiveness signature. The pro-invasive signature genes included 57 of 131 genes in the tumor proliferation signature meta-PCNA gene set (FET OR=36.8 and $p=5.3\times 10^{-57}$). We examined the impact of proliferation gene expression by clustering the 53 LUAD tumors using only the meta-PCNA signature genes; 10 of invasive tumors were clustered together with non-invasive tumors (SFig. 1e).

However, when we used our signature genes, excluding the meta-PCNA signature genes, the clustering result was completely agreed with the original clustering shown in Figure 1a (SFig. 1e), suggesting that tumor proliferation alone is not sufficient to explain the molecular differences between invasive and non-invasive tumors and our signature genes capture tumor invasive specific biologic features in addition to tumor proliferation features.”

3. Figure 2 shows samples in each independent primary tumor cohort that are classified into three groups (invasive, intermediate, and indolent). Classification based on IVS that is identified through the smallest and largest local minima is unusual and points to possible lack of robustness in the results. Supplementary Figure 2A indicates that histograms are quite noisy due to probably small sample size in several data sets. This can raise the concern of noise-driven cutoff selection and classification. Floating cut-off could bring bias in classification and influence survival analysis interpretation. The authors should perform such analysis with either fixed or standard percentile based cutoffs to demonstrate their findings are robust.

To assess the impact of floating cut-off vs. fixed cutoff in IVS based classification, we performed survival analysis using fixed absolute IVS value cutoff (0.05 vs. 0.95, 0.1 vs. 0.9, and 0.2 vs. 0.8) and fixed percentiles (top/bottom 5, 10, 20, and 30%). Patients grouped by fixed IVS cutoffs or percentiles had significantly different survival among/between groups in 6 out of 7 datasets (**Reviewer Table 1**). In the Wilkerson et al. dataset, which had the smallest sample size, the survival difference was not significant. The result suggests that both approaches are robust. When the dataset sample size is small, dataset-specific cutoff values may be more informative in stratifying patients into groups of different prognosis. The table is added as **Supplementary Table 6**.

Dataset	Group	Local minima	Fixed IVS value cutoff			Fixed IVS percentile cutoff			
			0.05, 0.95	0.1, 0.9	0.2, 0.8	5%	10%	20%	30%
Shedden et al. (n=371)	All three	0.0003	5.57E-06	2.75E-05	0.0007	0.03	9.40E-06	1.60E-05	5.66E-05
	Inv vs Ind	8.30E-05	3.70E-06	1.32E-05	0.0005	0.04	2.91E-05	2.86E-05	6.71E-05
TCGA LUAD (n=397)	All three	0.003	0.007	0.0009	0.03	0.006	0.02	0.0009	0.007
	Inv vs Ind	0.004	0.009	0.001	0.03	0.005	0.02	0.001	0.007
Okayama et al. (n=204)	All three	0.0002	0.0005	4.39E-05	2.34E-05	0.17	0.06	0.006	2.64E-05
	Inv vs Ind	0.0002	0.0005	2.80E-05	1.68E-05	0.09	0.01	0.007	5.25E-05
Tang et al. (n=111)	All three	0.007	0.02	0.0007	0.01	0.01	0.03	0.001	0.002
	Inv vs Ind	0.007	0.04	0.0006	0.01	0.01	0.007	0.0008	0.001
Der et al. (n=127)	All three	0.0002	2.81E-05	6.74E-05	6.28E-05	0.04	0.005	0.0009	8.38E-06
	Inv vs Ind	0.0003	1.69E-05	5.04E-05	4.94E-05	0.03	0.004	0.0004	1.01E-06
Rousseaux et al. (n=85)	All three	0.02	0.002	0.01	0.003	0.03	0.04	0.006	0.007
	Inv vs Ind	0.02	0.003	0.01	0.003	0.03	0.01	0.006	0.01
Wilkerson et al. (n=62)	All three	0.15	0.64	0.55	0.8	0.21	0.14	0.25	0.87
	Inv vs Ind	0.05	0.52	0.45	0.79	0.18	0.18	0.21	0.86

Reviewer Table 1. Survival differences among/between groups based on different IVS-based stratification schemes. For each dataset, tumors were separated into three groups (indolent, intermediate, and invasive) based on IVS. Log-rank test p-values were assessed for all three groups or the two extreme groups (invasive vs indolent).

In the revised manuscript, we added the following text on **page 9**:

Page 9: “To assess the robustness of survival association of IVS-based classification, we used multiple parametric and non-parametric IVS cutoff schemes to classify tumors. Both fixed IVS cutoff value and fixed IVS percentile-based classifications showed segregation of patient groups with significantly different survival (Supplementary Table 6).”

4. Although 70 cell lines were used for characterization of IVS, selected 5 cell lines from highly invasive group (NCI-H1373, SK-LU-1, NCI-H1792, A549, and NCI-H2009) all harbor mutant K-Ras. Whereas less invasive cell lines (HCC78, NCI-H3255, Calu3, HCC1833, HCC2279) are characterized by p53 and are KRAS wild-type. What is the selection criteria for cell lines to confirm invasiveness? Are highly scored cell lines without K-ras mutation (such as H1755, b-Raf mutant) associated with increased migration and invasion?

We thank the reviewer for this suggestion. The high IVS group contained cell lines with mutant KRAS (red dots in Fig. 2h) and wildtype KRAS (green dots in Fig. 2h). The cell lines initially used for the migration and invasion assays were randomly selected from stocks available to the lab. To address the reviewer question, additional experiments were performed to assess migration and invasion properties in three KRAS WT genotype invasive (high-IVS-scored) cell lines (H1755 as the reviewer suggested as well as H1975, and H1650). These cells showed similar migration and invasion activity as the 5 invasive KRAS mutant cell lines (**Reviewer Figure 5**), suggesting that KRAS mutation status is not sufficient to determine cell migration and invasion properties and that IVS identified cell lines with invasive activity, independent of KRAS mutation status.

Reviewer Figure 5. Migration and invasion assay in IVS predicted invasive and non-invasive cells. (Top) Representative images of migrated cells in transwell migration (left) or matrigel invasion (right) assay after 48 hours.

(Bottom) Quantification of migrated (left) or invaded cells through transwell matrigel at 48hr, 96hr, and 144 hr of 8 more invasive and 5 less invasive cell lines. Red indicates KRAS mutant high IVS cell lines, green indicates KRAS wildtype high IVS cell lines, and blue indicates low IVS cell lines.

We updated Fig. 2i-l and modified page 10 of the manuscript.

Page 10: "We selected 8 high IVS (ranked above median) with 5 KRAS mutant: H1373, SK-LU-1, H1792, A549, and H2009; 3 KRAS wild type: H1755, H1650 and H1975; and 5 low IVS (ranked below median):

HCC78, H3255, Calu3, HCC1833, HCC2279 cell lines. All 8 high IVS cell lines showed higher transwell migration and invasion within 48hr time period independent of their KRAS status compared to low IVS score cell lines (t-test p-values= 6.5×10^{-21} and 8.7×10^{-24} , respectively, Fig. 2i-j for migration and 2k-l for invasion), suggesting that the invasiveness phenotype predicted by our IVS-based ranking is significantly associated with experimentally observed biological phenotype and that KRAS mutation status is not a determining factor of invasiveness in these cells.”

5. A possible confounder in the analysis Fig 2b is the growth rate of the selected cell lines, and the focus on cell cycle kinases in the rest of the manuscript suggests this is a possibility. Is there difference in proliferation rate between Kras mutant cells (NCI-H1373, SK-LU-1, NCI-H1792, A549, and NCI-H2009) identified as highly invasive and K-Ras WT cells (HCC78, NCI-H3255, Calu3, HCC1833, HCC2279) classified as less invasive?

To distinguish between invasiveness and growth rate phenotype, we performed a growth rate assay from day 1 to day 5 for 7 highly invasive lines, (including 5 KRAS mutant (H1373, SK-LU-1, H1792, A549, and H2009) and 2 KRAS WT (H1650, H1975)) and 5 less invasive lines (HCC78, H3255, Calu3, HCC1833, HCC2279). Irrespective of classification as more invasive or less invasive, the growth rate difference between more vs. less invasive cells was not significant (the t-test p-values for days 2, 3, 4, and 5 were 0.13, 0.46, 0.86, and 0.75, respectively) (Reviewer Figure 6). These results suggest that the invasiveness activity of LUAD cells is independent of their basal proliferation rate. The figure is added as **Supplementary Figure 3b**.

Reviewer Figure 6. Growth rate for 7 more invasive LUAD cell lines (Top) and 5 less invasive cell lines (Bottom) from day 1 to day 5. KRAS wild type cell lines are marked by *.

This result is added on page 11.

Page 11: “To determine the potential effect of proliferation rate on invasiveness of these cell lines, we performed a growth rate assay to examine differences in growth rates between the cell lines predicted as more invasive versus less invasive. Using 7 more invasive cells (H1792, A549, H1975, H2009, H1650, H1373, and SK-LU-1) and 5 less invasive (H3255, HCC78, Calu3, HCC1833, and HCC2279), we observed no significant difference of growth rates between the two groups (t-test $p=0.13, 0.46, 0.86,$ and 0.75 for day 2, 3, 4 and 5, respectively, SFig. 3b). This suggests that the invasiveness activity of LUAD cells is independent of their basal proliferation rate.”

6. The authors focus on the role of AURKA and AURKB in lung tumor progression and speculate that they regulate cell migration and invasiveness in lung adenocarcinoma. The major concern is that the observed effect on cell migration and invasion assays might be due to reduction in cell proliferation and survival upon manipulation of AURKA and AURKB expression. Is the proliferation rate of cells affected by AURKA and AURKB downregulation? Nearly every published experiment of genetic knockout of either has significant effects on cell growth (e.g. see depmap database). The proliferation rate should be measured for all cell lines generated to estimate the effect of gene knockdown on cell proliferation and to clearly demonstrate migration/invasion effects are independent of proliferation

We appreciate the reviewer’s comment. To address this concern, we measured the effect of AURKA and AURKB knockout on cell proliferation from Day 1 to Day 8 in H1792 and A549 cells. We observed no significant difference in proliferation at day2 (48h), which is the time point used for the migration and invasion assays) (**Reviewer Figure 7**). As expected, we observed a decrease in cell proliferation in cells with deletion of AURKA and AURKB at later time points. These data suggest that the effect of AURKA and AURKB knockout on migration and invasion ability is not significantly affected by proliferation at the time point of this assay (48hr). This figure is added as Supplementary Figure 7d.

Reviewer Figure 7. Cell Proliferation shown (normalized to Day 1) showing effect of indicated sgRNA in (a) A549 and (b) H1792 cells from day 1 to Day 8.

We added the above results in the revised manuscript on page 14.

Page 14: “To examine whether the reduction in migration and invasion by CRISPR/Cas9 knockout of *AURKA* and *AURKB* is influenced by an effect on cell proliferation, we performed a proliferation assay for H1792 and A549 cells from Day 1 to Day 8. We observed no significant difference in proliferation at 48 hours after gene knockout, which is the time point at that migration and invasion were measured (SFig. 7d). As expected, we observed a difference in cell proliferation in knockout cells at later time points. Our proliferation assay data indicates that the effect seen in migration and invasion assay is not influenced by gene deletion effect on cell proliferation at the 48hr time point.”

7. It is known that KRAS positively regulates *AURKA* and *AURKB* expression in LUAD cell lines and Aurora kinase inhibition was suggested as a novel approach for KRAS-induced lung cancer therapy in 2016. More importantly, dual pharmacological inhibition of *AURKA* and *AURKB* reduced growth, viability, transformation, and induced apoptosis in vitro in an oncogenic KRAS-dependent manner including A549 cell line, that was also included in current study. Throughout the whole study only K-ras mutant cell lines were utilized to study the expression of *AURKA* and *AURKB* and the effect of their manipulation on different physiological outcomes including cell migration and invasion. By experimental design the invasive signature is defined regardless of mutation status. Moreover, Figure 2 shows that cell lines that are known by their wild type K-ras status are also assigned with high invasive score (for example H1755, PC19, H1975). The effect of *AURKA* and *AURKB* manipulation should be tested in KRAS wild type cell lines.

As suggested by the reviewer, we evaluated the effect of pan-aurora kinase inhibition in KRAS wild type cell lines. We assessed the effect of the aurora kinase inhibitor AMG900 on migration and invasion in H1975 and H1650 cell lines. As shown below (Reviewer Figure 8), AMG900 treatment significantly reduced migration and invasion in both KRAS wild type cell lines. This indicates that pan-aurora kinase inhibition effects on migration and invasion in these lung adenocarcinoma cells are not KRAS dependent. The figure is added as Supplementary Figure 8b.

Reviewer Figure 8. Migration and Invasion in 2 KRAS wildtype highly invasive cell lines H1650 and H1975 after treatment with AMG900 (0.1 μ M, 1 μ M). Representative images are shown at bottom. Data presented as mean \pm s.e.m. Significant comparison from two-way ANOVA test (n=3) is marked with asterisks (****: $p < 0.0001$).

We modified the revised manuscript text on page 15.

Page 15: "Migration and invasion were similarly reduced in invasive cells of wildtype KRAS (H1650 and H1975) after AMG900 treatment (two-way ANOVA test $p < 0.0001$ for dose of 0.1 μ M and 1 μ M, SFig. 8b), suggesting that pan-aurora kinase inhibition suppresses invasiveness in lung adenocarcinoma cells independent of KRAS driver mutation status."

8. Figure 6 and extended data figure 10 show clear differences in tumor burden and change between vehicle and AMG900 treated animals at several time points of tumor progression. This might be also explained by effect in tumor cell proliferation because at week 3 there are no visually detectable tumor sites. To neglect the effect of proliferation/survival on the tumor development in vivo (figure 6) and to clearly link the effect of AMG900 inhibitor with effect on invasiveness it will be informative to compare percentage of invasive tumors across mice that harbor matching tumor burden.

We thank the reviewer for this comment and suggestion. To control for the effect of proliferation in our in vivo mouse model, we compared invasive tumor lesions in subsets of mice with matching tumor burden. As shown in **Reviewer Figure 9**, invasive tumor lesions are reduced proportionally to overall size in AMG900 treated mice, suggesting that tumor invasion is impacted by aurora kinase inhibition independent of effect of tumor proliferation. This figure is included as **Supplementary Figure 10d** and text is added on page 18.

Reviewer Figure 9. Stacked plot showing invasive tumor lesion per total tumor area in vehicle and AMG900 treated animals with equal overall tumor burden at week 9, 13 and 17 (n=2).

Page 18: “Because total tumor size is associated with proliferation, we performed analysis of invasion extent in animals bearing similar tumor burdens. Histopathological data from animals in the vehicle and AMG900 treatment group showed a significant reduction in invasive tumor with aurora kinase inhibition, further supporting a direct link between aurora kinase inhibition and tumor invasion phenotype independent of total tumor size (SFig. 10d).”

Minor Concerns:

1) Data visualization requires improvement. To simplify interpretation, and to visualize patterns in the complex network, networks might be grouped in according to their link to inhibitor term or GO term. Extended data figure 4 does not show any useful information and requires significant improvement.

We thank the reviewer for the suggestion. We modified **Supplementary Figure 4** with a subnetwork focusing on pro-invasive and indolence signatures (**Reviewer Figure 10**). We overlaid enriched gene sets for; 1) breast cancer invasive signatures, 2) G2M, EMT, and MTORC1 pathways from HALLMARK databases, 3) lung cancer survival signatures; and 4) curated tumor suppressor genes. This revised figure provides a clear view of how the signatures are associated with individual gene sets. In addition, there are distinct enrichments of TPX2/AURKB and COL1A2 subnetworks for multiple poor prognostic signatures.

Reviewer Figure 10. Subnetwork views focusing on pro-invasive and indolence signature genes. Nodes are colored for the pro-invasive (red) and indolence (green) signature genes on the top-left corner. Genes from the selected gene sets from Figure 1d are projected on the network and colored according to their association with the tumor invasion.

2) P values might be displayed in the figures, and statistical analysis at least mentioned in the figure legends. Figure 4E should contain statistical data. For example, cells containing statistically significant change can be labeled with (*)

We added values from a generalized linear regression model; $\text{lm}(\text{phenotype} \sim \text{dose})$ in Supplementary Table 10. We add asterisks to indicate p-values of the fit line in a dose dependent manner in Figure 4e. We also labeled p values in Supplementary Table 10 for migration, invasion, and wound healing assays for all 5 invasive cell lines in Supplementary Figure 8a.

Reviewer #2 (Remarks to the Author): Expert in lung cancer therapy and mouse models

Title: Integrative Network Analysis of Early-Stage Lung Adenocarcinoma Identifies Aurora Kinase Inhibition as Interceptor of Invasive and Progression

Summary: The authors have created a computational pipeline analyzing their in-house RNA-seq of early-stage lung adenocarcinoma (n = 53) to determine the signatures associated with invasive and indolent groups. The signature gene sets were validated by the expression data of the *Kras*^{+/-}*Tgfbr2*^{-/-} genetically engineering mouse model. They established the Invasiveness score (IVS) based on these signature genes to predict prognosis of stage I lung adenocarcinoma (LUAD) patients in 7 published available cohorts. Aurora kinases (AURKA and AURKB) were identified as regulators of the pro-invasive network and validated as highly expressed by tissue microarray. The authors demonstrated that Aurora kinase inhibitor, AMG900, reduced the migration and invasiveness of human cell lines with high IVS and the invasive mouse model.

The authors analyzed multiple data sets, including human, mouse model and cell lines, to support their findings. This is an important area of investigation of potential clinical relevance. However, the finding of Aurora kinases as key regulators has been demonstrated in multiple lung cancer studies (summarized in PMID: 33202573).

We thank the reviewer for the encouraging comments. With regards to the existing literature on Aurora kinase, most studies focus on proliferation. Our study focuses on Aurora kinases' effect on cell invasion independent of its role in regulating cell proliferation.

Major comments

1. In line 125-126, the authors claim that their signatures genes based on clustering approach are “more informative” than comparisons based on histologically defined groups. The comparison between two histology groups defines 793 DEGs, while the clustering approach provides 1322 genes, representing a 40% increase in genes. The higher number of candidates often improves the prediction but may also increase the chance of false positives and overfitting. To demonstrate that the clustering approach provides a gene set with more biological meaning than phenotype-comparisons, authors should build the IVS based on the same number of genes for each approach and then compare the predictions.

We thank the reviewer for the suggestion to compare the survival associations of the signatures with IVS using the same number of genes. We balanced the number of genes by selecting the top 793 genes from our signature genes. IVS was calculated using the two signatures for the 4 largest LUAD datasets (Shedden,

TCGA, Okayama, and Tang et al.). Tumors from each dataset were classified into three groups based on local minima of IVS as described in Methods, and survival differences were assessed with LRT p-values (**Reviewer Figure 11**). Due to the large number of overlapping genes, it is not surprising that survival differences among groups based on both signatures were similar. However, expression-driven clustering DEGs provided slightly better results for Okayama and Tang et al. (LRT p-values = 9.1×10^{-5} vs 0.0003 for Okayama et al., 0.004 vs 0.05 for Tang et al. based on expression-driven clustering and histology-based DEGs, respectively).

As described in Response 1 to Reviewer 1, compared to histology-based signatures, the expression differences of the overlapping genes between the signatures by the two approaches were more significant in the expression-driven approach (**Reviewer Figure 1**) and expression-driven clustering approach captures unique molecular features related to tumor invasiveness compared to histology- or node metastasis status-based classification (**Reviewer Figures 2-3**).

Reviewer Figure 11. Stratification of tumors into three groups based on IVS. Invasive (high IVS), intermediate (middle IVS), and indolent (low IVS) tumors determined based on IVS. IVS was measured using the top 793 DEGs by expression-driven clustering (top panel) or histology-based classification (bottom panel). Five-year survival probability was shown in a KM curve with corresponding LRT p-values for three groups or two groups (invasive vs indolent).

2. The authors utilized *Kras*^{+/−}*Tgfbr2*^{−/−} genetically engineering mice to validate pro-invasive signature genes identified in human data and access the expression changes induced by the aurora kinase inhibitor. However, in the original study of the mouse model, Borczuk and colleagues (Ref 32) claim that *Tgfbr2* deletion changed the tumor microenvironment and induced an invasive phenotype. Therefore, the authors should discuss if the human data showed deregulation of the TGFBR pathway or were enriched by DEGs associated with changes in the tumor environment based on the mouse model.

We appreciate the reviewer's comment. Indeed, *TGFBR2* was downregulated in invasive tumors compared to indolent tumors in the human adenocarcinoma RNAseq dataset (FDR=0.0016). In addition,

indolence signature genes were significantly enriched for genes potentially responsive to TGFB (FET OR= 7.3 $p= 2.4 \times 10^{-6}$ against GO_TRANSFORMING_GROWTH_FACTOR_BETA_RECEPTOR_BINDING, Supplementary Table 3).

As suggested by the reviewer, we investigated gene expression changes in the *Tgfr2* knockout mouse model (GSE27675). Upregulated genes (T-test $p < 0.01$ & $FC > 2$) in *Tgfr2* knockout stroma significantly overlapped with pro-invasive signature genes (FET OR= 4.12, p -value= 4.3×10^{-5}). The significant overlap was observed only with the COL1A2 subnetwork (FET OR=9.7, $p=6.1 \times 10^{-5}$) but not with the TPX2/AURKB subnetwork (FET OR= 1.2 and $p=0.57$). The downregulated genes in stroma also overlapped with the indolent signature genes (FET OR= 3.0 $p=0.05$). This suggests that both tumor intrinsic molecular alterations and the tumor microenvironment contribute to tumor invasion phenotype as well as to invasion signatures. We added these results on pages 18 -19 and a brief comment in the Discussion section of the revised Manuscript on pages 21 - 22.

Page 18-19: "In the human LUAD dataset, TGFBR2 was included in the indolence signature and the indolence signature genes were significantly enriched for genes involved in the TGFB pathway (Supplementary Table 3), confirming the importance of TGFBR2 in tumor invasion and providing a potential model for tumor cell aurora kinase signaling impact on the tumor microenvironment. In human cancer cells, RNAseq data showed that aurora kinase inhibition suppressed the pro-invasive but not the indolence signature genes. When we compared differentially expressed genes between the mouse model and human data, genes upregulated in *Tgfr2* knockout mouse stroma were significantly enriched for the COL1A2 subnetwork (FET OR= 9.7 and $p=6.1 \times 10^{-5}$) but not for the TPX2/AURKB subnetwork (FET OR=1.2 and $p= 0.57$). In addition, AMG900 treatment in cancer cells suppressed most of the genes in the TPX2/AURKB subnetwork but not in the COL1A2 subnetwork. This indicates that expression changes of the TPX2/AURKB subnetwork likely reflects tumor intrinsic features while those of the COL1A2 subnetwork likely captures features of stromal cells in tumor microenvironment impacted by alterations in tumor cell aurora kinase signaling.

Pages 21-22: "Using a genetically engineered mouse model, we previously reported *Tgfr2* as a determinant of invasive phenotype of early-stage lung adenocarcinoma, likely through modulating the tumor microenvironment that was enriched for expression of genes in the COL1A2 subnetwork and not for genes in the TPX2/AURKB signaling subnetwork. It is important to note that tumor invasion phenotype in tumor tissue is complex depending on both tumor intrinsic molecular features and interactions between tumor cells and tumor microenvironment that may be mediated through aurora kinase signaling. Further investigations including single cell genomic studies will be necessary to elucidate how the tumor invasion phenotype is orchestrated by these subnetworks in a complex, interactive, and heterogeneous cellular environment."

3. It has been shown that KRAS positively regulates AURKA expression (PMID: 26842935). Both cell lines A549 and H1792 utilized to evaluate aurora kinase inhibitors have KRAS mutation. The authors should address the relationship between AURKA/B and driver gene mutations. If the high expression of AURA/B in the invasive group is independent of mutations in RAS pathways, they should evaluate the therapy in other human cell lines without KRAS mutations. A549 also has an inactivating LKB1 co-mutation. Given the previously published relationship with Aurora Kinase and LKB1 (PMID: 28967900), the authors should include discussion of its potential role.

We have addressed this important issue in responses 4 and 7 to reviewer #1. **Reviewer Figure 5** indicates

that KRAS mutation status is not a determining factor for cell invasiveness. We evaluated the effect of pan-aurora kinase inhibition in KRAS wildtype lung adenocarcinoma cell lines H1650 and H1975. As shown in **Reviewer Figure 8**, migration and invasion were significantly reduced in both wild type KRAS cell lines, suggesting that KRAS mutation status is not a determining factor for cell response to pan-aurora kinase inhibition.

We appreciate the reviewer’s comment about the relationship between Aurora kinase and LKB1 activation. We have addressed this in the discussion section on page 21.

Page 21: “A recent study reported the role of driver mutation LKB1 and AURKA mediated phosphorylation of LKB1 in proliferation, invasion and migration of NSCLC cells. Among our panel of 8 highly invasive cell lines, only A549 harbors a LKB1 nonsense mutation and shows similar results with other LKB1 WT invasive cells from our CRISPR mediated knockout as well as pan-inhibition of Aurora kinase A and B, suggesting that the reduction observed in migration and invasion by deleting *AURKA* and *AURKB* is independent of LKB1 driver mutation status.”

4. The authors claim the AMG900, aurora kinase inhibitor, impairs AKT/mTOR and EMT pathways. However, they did not discuss or demonstrate if the impact was via aurora-kinase dependent or independent pathways. Were the AKT/mTOR and EMT markers also changed in A549 and H1792 with CRISPR/Cas9 deletion *AURKA*/*AURKB*?

As suggested by the reviewer, we tested the effect of CRISPR/Cas9 mediated *AURKA* and *AURKB* knockout on AKT/mTOR and EMT signaling in H1792 and A549 cells. Immunoblot shows that double knockout of both *AURKA* and *AURKB* leads to diminished expression of pAKT, p-mTOR and EMT markers in both cell lines (**Reviewer Figure 12**). This suggests that *AURKA* and *AURKB* together mediate invasiveness through regulation of AKT/mTOR and EMT signaling. The figure is updated as Figure 5e and Supplementary Figure 9e and text is modified on page 17.

Reviewer Figure 12. Western blot for indicated proteins in H1792 and A549 cells treated with indicated sgRNAs

Page 17: “To further examine the effect of *AURKA* and *AURKB* knockout on AKT/mTOR and EMT signaling, we examined protein expression for both pathways in CRISPR knockout A549 (Fig. 5e) and H1792 (SFig. 9e) cells. p-mTOR and p-AKT were suppressed in cells deleted for both *AURKA* and *AURKB*, but not with single deletion. Similarly, EMT marker N-Cadherin, Claudin 1, Vimentin and Snail were suppressed only in cells with double deletion of *AURKA* and *AURKB* (Fig. 5e and SFig. 9e). Together, these data suggest that AKT/mTOR and EMT pathway activation in invasive lung adenocarcinoma cells is dependent upon aurora kinase signaling.”

5. In figure 3e, the survival curve of patients with highest *AURKA* score (score =2) was not distinguished from the lowest score. How are the KM curves based on two groups (0 vs. > 0)? What is the outcome based on *AURKB*?

We thank the reviewer for the suggestion. We separated patients into two groups (score = 0 vs score = 1+), and patients within the two groups showed a significant difference in 5 year overall survival (LRT $p=0.0004$, **Reviewer Figure 13** below). Unfortunately, we don't have enough samples with positive *AURKB* staining to compare patients' survival based on *AURKB* (# of samples with survival and staining: 251 for *AURKA* and 62 for *AURKB*).

Response Figure 13. KM plot of LUAD groups separated by *AURKA* score.

6. Aurora kinases are known to be associated with cell proliferation. The growth rate or the doubling time of a cell line will depend on its aurora kinase level. The migration and invasive assay are at least 48 hours in duration, which is longer than the doubling time of the cell lines utilized (e.g., 22-24hrs for A549).

Therefore, the authors should shorten the timeframe ($t = 24\text{hr}$) for these assays to eliminate the effect of cell proliferation following migration through the membrane.

We agree that it is important to demonstrate the effect of AURKA and AURKB knockout on cell proliferation so as to distinguish the effect on migration and invasion phenotypes from proliferation. To address this concern, we measured the effect of AURKA and AURKB knockout on cell proliferation at the same time point, 48hr that was used for Migration and Invasion assays in Fig. 4c and d. We do not observe differences in cell proliferation for up to 48 hours in both H1792 and A549 cells (**Reviewer Figure 7**). After 48 hours, as expected because of the dependence of these cells on AURKA and AURKB, the proliferation rate for the knockdown cells was reduced.

To specifically address the reviewer's suggestion, we performed migration and invasion assays at 24 hours for 2 representative cell lines A549 (doubling time= 24hr) and H2009 (doubling time=29.8hr), which is shorter than the doubling time of both cell lines. Migration and invasion were significantly reduced in both cell lines at 24hr assay time with AMG900 treatment (**Reviewer Figure 14**), which suggests that migration and invasion reduction by pan-Aurora kinase inhibition are independent of cell proliferation at these time points. This figure is added as **Supplementary Fig 8c** and text is updated on *page 15*.

Reviewer Figure 14. Migration and Invasion from transwell migration and invasion assays in 2 KRAS mutant highly invasive cell lines A549 and H2009 treated with AMG900 (0.1μM, 1μM), measured at 24hr (top). Representative images are shown at bottom. Data presented as mean ± s.e.m. Significant comparison from two-way ANOVA test (n=3) is marked with asterisks.

Page 15: “Cell line growth rates can affect the readout of migration and invasion assays. When migration and invasion assays were performed at 24 hours, which is shorter than doubling intervals of A549 and H2009 cells, we observed a similar impact by aurora kinase inhibition as in experiments performed at 48 hours (SFig. 8c). This suggests that invasion and migration are regulated by aurora kinase signaling independent of effects on proliferation.”

Minor comments

7. The methods (line 616-619) indicate that AURKA and AURKB were scored differently based on their nuclear or cytoplasmic location in the TMA data analysis. The authors should explain the biological meaning for the deviation of the scoring system.

Prior publications (PMIDs: 23411487, 17121938) indicate certainty in nuclear localization for both proteins and suggestion of cytosolic staining for AURKA. While there is data that AURKB has changing subcellular localization based on interaction with p53, cytosolic staining was not seen in our cases. Therefore, to best interpret both protein staining patterns, nuclear staining was required for both analyses. Since no cytosolic staining was seen for AURKB and was scored 0 in all cases, it had no impact on H-score. The consistent presence of both nuclear and cytoplasmic staining for AURKA in these cases was seen and therefore both were included in the H-score. We added these explanations in the Methods section on *pages 30 - 31*.

Pages 30-31: "A previous study indicates certainty in nuclear localization for both proteins and suggestion of cytosolic staining for AURKA. While there is data that AURKB has changing subcellular localization based on interaction with p53, cytosolic staining was not seen in our cases. Therefore, to best interpret both protein staining patterns, nuclear staining was required for both analyses. Since no cytosolic staining was seen for AURKB and was scored 0 in all cases it had no impact on H-score. The consistent presence of both nuclear and cytoplasmic staining for AURKA in these cases was seen and therefore both were included in the H-score."

Reviewer #3 (Remarks to the Author): Expert in lung cancer clinical research, lung cancer invasion and genomics

The paper from Seungyeul Yoo et al identifies aurora kinase as one of key regulators of invasiveness in early stage lung adenocarcinoma and inhibition of aurora kinases attenuate tumor invasion both in vitro and in vivo. Collectively, authors' key findings include: a) generate gene signatures which distinguish invasive and indolent lung adenocarcinoma and validated the functions of these gene signatures in independent cohorts, human tissues, cell lines and genetic mouse models. b) identifies aurora kinase inhibition as therapeutic approaches to block cancer invasion and progression in cells and animal models. The authors make a great effort to distinguish patients with high invasive properties even at early stages and provides potential therapeutic targets for treatment, which fills the gap and be of great interest to the clinicians in this field. However, several terms are not very convincing which need to be addressed in the revised version.

1.Line 106, about the definition of "invasive" and "indolent". Here, two distinct groups were identified based the clusters. Cluster in group with pathologically aggressive subtype was defined as "invasive", cluster in group with pathologically less invasive subtypes was defined as "indolent". If this definition was finally based on pathological classification, then why not do differential cluster analysis according to pathological classification first. Again, in line 268, categorizing tumors into indolent and invasive groups was based on their histology which is not consistent with first one. Different classification of "invasive" and "indolent" were applied in this paper. It need to clarified.

We agree with the reviewer's point that the terminology of "invasive" and "indolent" tumor groups or signature genes could be misleading and should be clarified. For the 53 original RNAseq, tumor samples were first classified into two groups based on gene expression-driven unsupervised clustering and each group was labeled based on enriched histopathological subtypes. While this classification was consistent with the histopathological information for most samples, 3 histologically invasive and 4 histologically non-invasive tumors were classified in the opposite groups based on unsupervised clustering (Figure 1a and 1b). As the reviewer pointed out, we compared our signatures from expression-driven clustering with gene signatures by histology-based classification and showed that our classification resulted in a larger number of signature genes capturing more distinctive molecular features between invasive and non-invasive tumors (please see responses to Reviewer 1 comment #1 and Reviewer 2 comment #1).

We have clarified the terminology used for tumors and signature genes. For the tumors, we use the terms "Invasive" and "non-invasive" for assignment based on gene expression. For the signature gene sets, we use the terms "pro-invasive" and "indolence". For the patients stratified based on IVS, we grouped them into "Invasive", "Intermediate", and "Indolent" groups associated with clinical outcomes. We updated the revised manuscript based on this change.

For the tissue microarray data described on line 399, analysis was based on classification by histology. We agree this could lead to confusion for the reader, so we changed labels of tumor groups to "invasive_histology" for AC, PAP, MP, and SOL tumors and "non-invasive_histology" for MIA, AIS, LPA tumors on page 13 and Figure 3d.

2.Line 93, 53 early stage lung adenocarcinoma samples were analyzed. How about driver gene profile for those patients? Cancer cells with different driver gene profile have quite distinct biological characteristics which might contribute to the invasiveness.

We appreciate the reviewer's suggestion. Mutation status of lung cancer driver genes (EGFR, KRAS, and TP53) were profiled through a targeted sequencing using Illumina Truseq 48 Cancer Gene Panel for the 53 early-stage lung adenocarcinoma (**Reviewer Figure 15**). EGFR mutation was enriched within non-invasive tumors (chi-square test $p=0.003$). The opposite trend was observed for TP53 mutation but the p-value was not significant (chi-square test $p= 0.07$). KRAS mutation was observed in both groups (Chi-square test $p= 0.33$).

Reviewer Figure 15. Mutational landscape of lung cancer driver genes among 53 early-stage lung adenocarcinoma.

We added the figure as Supplementary Figure 1b and added text on page 5.

page 5: “EGFR mutation was enriched within the non-invasive tumors classified by the signature (chi-square test $p=0.003$, SFig 1b) while KRAS and TP53 mutation were observed in both invasive and non-invasive tumors (chi-square test $p= 0.33$ and 0.07 for KRAS and TP53, respectively, SFig 1b).”

3.Line 172. In this paragraph, gene signature was further validated in 7 databases by overall survival analysis. The association of gene signature with survival should also be analyzed in his own patient group.

Unfortunately, we do not have access to the survival outcome information for the 53 patients.

4.Line 71, type 2 TGF Beta Receptor was identified as determinant of invasiveness and metastasis of localized lung adenocarcinoma. In current study, TPX2 and AURKB were found as key regulator of pro-invasiveness. Then, how to explain different findings?

The tumor invasive phenotype is complex, depending on both tumor intrinsic molecular features and tumor microenvironment. As the reviewer pointed out, we previously reported TGFBR2 as a determinant of invasive phenotype of early-stage lung adenocarcinoma, likely through modulating the tumor microenvironment (PMID:21911454). In this study, we used a causal regulatory network using stage I TCGA LUAD samples and identified TPX2/AURKB as key regulators of tumor invasion, likely through modulating tumor cell intrinsic molecular features.

Invasive tumors induced by tumor intrinsic change or tumor microenvironment influences may share common molecular features. Indeed, TGFBR2 is one of the indolence signatures (FDR=0.0016) and the indolence signature genes were significantly enriched for a TGFBR gene set (FET OR= 7.3 $p= 2.4 \times 10^{-6}$, Supplementary Table 3). In addition, a subnetwork surrounding TGFBR2 was enriched for the indolence signature (**Reviewer Figure 16**, FET OR=21.5 and $p=1.3 \times 10^{-5}$). Moreover, our signature successfully classified invasive and non-invasive tumors from *Kras^{+/+}Tgfbr2^{-/-}* mouse models (Figure 1e). These confirm the important role of TGFBR2 in tumor invasion.

When we investigated expression changes by *Tgfbr2* knockout in the mouse model, the upregulated stromal cell genes significantly overlapped with the COL1A2 subnetwork (FET $p= 6.1 \times 10^{-5}$) but not with the TPX2/AURKB subnetwork (FET $p=0.57$). RNAseq data using human cancer cells showed that the aurora kinase inhibition suppressed the pro-invasive genes (FET $p = 7.7 \times 10^{-62}$ and 7.2×10^{-10} for A549 and H1792 cells, respectively) but not the indolence signatures (FET $p=0.32$ and 0.98 for A549 and H1792) suggesting that molecular mechanisms underlying the indolence genes may be independent from direct aurora kinase regulation of tumor cells. These results suggest that there are multiple paths inducing tumor invasiveness involving interactions between tumor cells and the microenvironment.

Reviewer Figure 16. A subnetwork surrounding TGFBR2 within 2 layers. Indolence signature genes were colored in green.

We modified the Result on pages 18 -19 and the Discussion on page 21.

Pages 18-19: “In the human LUAD dataset, TGFBR2 was included in the indolence signature and the indolence signature genes were significantly enriched for genes involved in the TGFBR pathway (Supplementary Table 3), confirming the importance of TGFBR2 in tumor invasion and providing a potential model for assessing tumor cell aurora kinase signaling impact on the tumor microenvironment.”

Page 21: “Using a genetically engineered mouse model, we previously reported *Tgfbr2* as a determinant of invasive phenotype of early-stage lung adenocarcinoma, likely through modulating tumor

microenvironments that was enriched for expression of genes in the COL1A2 subnetwork and not for genes in the TPX2/AURKB signaling subnetwork.”

5.Line 327, DEGs were highly consistent in two cell lines treated with aurora inhibitors. Then how about consistence of these DEGs with gene cluster identified in figure 1?

The genes downregulated by AMG900 treatment were enriched for the invasive signatures in both cell lines (FET $p = 7.7 \times 10^{-62}$ and 7.2×10^{-10} for A549 and H1792 cells, respectively) but upregulated genes did not significantly overlap with the indolence signature (FET $p=0.32$ and 0.98 for A549 and H1792). This suggests that aurora kinase activity directly regulates only the invasive signature genes in tumor cells. We performed functional analysis of the downregulated genes with HALLMARK gene sets as reported in Supplementary Table 12. These results were summarized in the manuscript on *pages 16 - 17*.

Additionally, we compared the AMG900 treatment downregulated genes with the gene sets listed in Figure 1d. As shown in **Reviewer Table 2**, gene sets enriched with pro-invasive signature genes were suppressed by AMG900 treatment in both A549 and H1792 cells.

C2 genesets	A549			H1792		
	Overlapping genes	OR	p-value	Overlapping genes	OR	p-value
SHEDDEN_LUNG_CANCER_POOR_SURVIVAL_A6	247	13.39	1.95E-135	183	5.6	1.83E-57
BIDUS_METASTASIS_UP	139	20.02	3.81E-90	119	10.28	7.14E-56
WINNEPENINCKX_MELANOMA_METASTASIS_UP	107	21.57	1.69E-71	60	4.71	1.43E-17
SARRIO_EPITHELIAL_MESENCHYMAL_TRANSITION_UP	111	17.34	7.01E-69	78	6.16	1.48E-27
VANTVEER_BREAST_CANCER_METASTASIS_DN	63	12.23	1.04E-34	52	6.46	1.50E-19
LIAO_METASTASIS	128	3.22	3.00E-24	116	2.2	5.31E-12
PUIFFE_INVASION_INHIBITED_BY_ASCITES_UP	40	8.98	1.58E-19	30	4.27	7.87E-09
POOLA_INVASIVE_BREAST_CANCER_UP	62	2.83	8.31E-11	36	1.14	0.263080918
GOTZMANN_EPITHELIAL_TO_MESENCHYMAL_TRANSITION_UP	25	5.45	1.77E-09	23	3.78	2.10E-06
JAEGER_METASTASIS_UP	19	7.41	4.31E-09	14	3.56	0.000321038
ALONSO_METASTASIS_EMT_UP	15	6.67	4.73E-07	18	7.39	2.61E-08
ZHONG_SECRETOME_OF_LUNG_CANCER_AND_ENDOTHELIUM	19	3.86	8.18E-06	18	2.82	0.00046473
LI_AMPLIFIED_IN_LUNG_CANCER	35	2.36	2.47E-05	36	1.93	0.000706096
SWEET_LUNG_CANCER_KRAS_DN	66	1.74	7.82E-05	92	2.07	7.94E-09
WU_CELL_MIGRATION	27	1.64	0.016185449	42	2.24	1.71E-05
PICCALUGA_ANGIOIMMUNOBLASTIC_LYMPHOMA_UP	27	1.46	0.04713467	53	2.7	1.23E-08
SCHUETZ_BREAST_CANCER_DUCTAL_INVASIVE_UP	39	1.21	0.151249157	64	1.72	0.000145364
ANASTASSIOU_MULTICANCER_INVASIVENESS_SIGNATURE	5	0.8	0.744413926	16	2.51	0.002530136

Reviewer Table 2. Functional enrichment analysis of genes down-regulated by AMG900 treatment within gene sets identified in Figure 1d. Association was tested via Fisher’s exact test and significant p-value <0.001 is noted in yellow.

6.Line 337 and line 388, In cell lines, no significant changes of COL1A2 subnetwork were found, while in animal models, obvious changes of collagen were found after treatment of AMG900. How to explain this? At least, it should be explained in the discussion part.

We appreciate the reviewer’s comment. Both tumor intrinsic molecular features and tumor microenvironment influences contributed to tumor invasion signatures derived from bulk tissue profiles. As shown above in the response to the Reviewer 2 Comment #2, the upregulated genes by Tgfr2 knockout in mouse stroma were significantly enriched for the COL1A2 subnetwork (FET OR=9.7, $p=6.1 \times 10^{-5}$) but not for the TPX2/AURKB subnetwork (FET OR=1.2, $p=0.57$).

In cell lines, AMG900 treatment suppressed most of genes in the TPX2/AURKB subnetwork (which likely are tumor intrinsic features) but not genes in the COL1A2 subnetwork (which likely reflect features of stromal cells in tumor microenvironment). In mouse models, tumor cells interact with cells in the tumor microenvironment. Molecular changes in tumor cells may induce molecular changes in cells in the tumor microenvironment. This may explain why AMG900 treatment in mice induced changes of collagen.

We added the results on page 19 and the following statements in Discussions of the revised manuscript on pages 21 - 22.

Page 19: “When we compared differentially expressed genes between the mouse model and human data, upregulated genes by *Tgfr2* knockout in mouse stroma were significantly enriched for the COL1A2 subnetwork (FET OR= 9.7 and $p=6.1\times 10^{-5}$) but not for the TPX2/AURKB subnetwork (FET OR=1.2 and $p=0.57$). In addition, AMG900 treatment in cancer cells suppressed most of the genes in the TPX2/AURKB subnetwork but not in the COL1A2 subnetwork. This indicates that expression changes of the TPX2/AURKB subnetwork likely reflects tumor intrinsic features while those of the COL1A2 subnetwork likely captures features of stromal cells in tumor microenvironment impacted by alterations in tumor cell aurora kinase signaling.”

Pages 21-22: “It is important to note that tumor invasion phenotype in tumor tissue is complex depending on both tumor intrinsic molecular features and tumor microenvironment that may be mediated through aurora kinase signaling. Further investigations including single cell genomic studies will be necessary to elucidate how the tumor invasion phenotype is orchestrated by these subnetworks in a complex, interactive, and heterogeneous cellular environment.”

7.Line 387, inhibition of Aurora with AMG900 were associated with neo-angiogenesis. The gene signatures were associated with cell cycle, EMT and angiogenesis when compared with Hallmark gene sets (paragraph 2, line 142). Thus, it is better to have more analysis including cell cycle and EMT changes after treatment of AMG900 in animal models?

To broaden the analysis of cell cycle and EMT changes after AMG900 treatment, we performed immunohistochemistry for E-cadherin, pAKT in murine tumors. AMG900 treatment was associated with increased expression of E-Cadherin and decreased expression of pAKT (**Reviewer figure 17**). These results were added to Supplementary Fig 11 and on pages 19 - 20.

Reviewer Figure 17. IHC for E-Cadherin and pAKT in *Kras(G12D)/Tgfr2^{-/-}* mouse model with AMG900 treatment. (a-b) E-Cadherin immunostaining showing reduction in staining intensity in invasive tumors areas in (a) vehicle controls when compared to uniform strong membranous staining in (b) AMG900 treated mice. c) Box plot shows values of twelve regions of interest for E-Cadherin staining analyzed by Image J plugin IHC profiler, $p=0.004$. (d-e) Immunohistochemistry for pAKT showing strong staining in tumors cells in (d) vehicle animals while significantly low staining in (e) AMG900 treated mice. f) Box plot shows values of twelve regions of interest for pAKT staining analyzed by Image J plugin IHC profiler, $p=0.01$. (A, B, D, E, Original magnification x150).

Pages 19-20: “We observed a loss in E-Cadherin expression in vehicle treated invasive tumors, while AMG900 treated animals showed uniform strong expression of E-Cadherin ($p=0.004$, SFig. 11a-c). Immunohistochemical staining for pAKT showed a strong positive staining in vehicle animal tumors, while loss was observed in AMG900 treated animals ($P=0.01$, SFig. 11d-f).”

8. Line 415, small spelling mistake: squamous cell carcinoma.

Thank you. We corrected the typo in the manuscript on page 21.

REVIEWER COMMENTS

Reviewer #2 (Remarks to the Author):

The authors have addressed the reviewers' concerns.

Reviewer #3 (Remarks to the Author):

The authors replied all concerns by 3 reviewers point by point and make the revised version more acceptable.

I am quite sure current paper could provide a solid scientific foundation for future clinical trials based on reported gene signature and potential application of aurora kinase inhibitors.

Reviewer #4 (Remarks to the Author): Expert in lung cancer genomics and signalling

Comments on the authors' response to Reviewer 1's comments

Major concerns:

Point 1.

The author provided evidence supporting the gene signature-based classification as an alternative classifier for tumor invasiveness than histology alone. Although the improvement, to my opinion, is relatively small and not striking, mainly because 1) histology classification was already very good and 2) the authors derived their gene expression signatures from tumor classified based on histology staging. So, I feel the signature is valid, albeit one among many published metastatic signatures. And practically such signature is unlikely to see clinical application or replace histological classification.

Point 2.

The authors have addressed the reviewer's concern. They removed proliferation associated genes and showed the remainder of the signature can still classify the tumors properly.

Point 3.

The authors partially address the reviewer's question. Local minima can lead noise-driven cut-offs. In Figure 2a-g, the authors should analyze their data with either top vs. bottom quartiles or divide each patient cohort evenly into 3 groups for this analysis. I understand that not every dataset would show significance, but this would be a more robust approach.

Point 4.

The authors have adequately addressed this question. The invasive phenotype does not appear to be KRAS dependent.

Point 5.

The authors have adequately addressed this question. The invasive phenotype does not appear to be proliferation driven.

Point 6.

The authors partially address the reviewer's concern. They showed that the invasion/migration phenotype occurred within 2 days prior to the proliferation slowdown of AURKA and AURKB KO cells. It is nevertheless surprising that these stable AURKA and AURKB CRISPR KO cells are viable at all and can continue to proliferate. Unless the CRISPR KO cells are actually hemizygous KO, it would be far more interesting to understand how these KO cells can transit through mitosis in the absence of AURKA or AURKB without massive chromosomal instability.

Point 7.

The authors have done experiments to address the reviewer's concern. But the migration vs. proliferation question remains.

Point 8.

The authors have partially addressed the reviewer's question. The plot has $n=2$ so it is not clear if the difference is statistically significant.

Reviewer #4 (Remarks to the Author): Expert in lung cancer genomics and signaling

Comments on the authors' response to Reviewer 1's comments

Major concerns:

Point 1.

The author provided evidence supporting the gene signature-based classification as an alternative classifier for tumor invasiveness than histology alone. Although the improvement, to my opinion, is relatively small and not striking, mainly because 1) histology classification was already very good and 2) the authors derived their gene expression signatures from tumor classified based on histology staging. So, I feel the signature is valid, albeit one among many published metastatic signatures. And practically such signature is unlikely to see clinical application or replace histological classification.

We appreciate the reviewer's opinion supporting the validity of the gene signature. We view the signature as being complementary to histology, the value of this has been demonstrated in other solid tumors such as breast cancer. Importantly, our gene signatures revealed potential molecular mechanisms/regulations underlying tumor invasiveness so that drugs targeting these pathways can be identified and tested.

Point 2.

The authors have addressed the reviewer's concern. They removed proliferation associated genes and showed the remainder of the signature can still classify the tumors properly.

Point 3.

The authors partially address the reviewer's question. Local minima can lead noise-driven cut-offs. In Figure 2a-g, the authors should analyze their data with either top vs. bottom quartiles or divide each patient cohort evenly into 3 groups for this analysis. I understand that not every dataset would show significance, but this would be a more robust approach.

We performed additional analyses following the reviewer's suggestion. Splits based on parametric and non-parametric cutoffs suggested by reviewers 1 and 4 resulted in significant log-rank test p-values in all datasets except for the Wilkerson et al. dataset which had the smallest sample size (Supplementary Figure 3 and Supplementary Table 6), suggesting that the IVS is robust for ranking tumor's invasiveness risks. The results of classifying patients into 3 or 4 groups of equal size are shown below (Reviewer Figure 1). The figure was added as Supplementary Figure 3.

P-values of log-rank test depend on both samples size of each group and hazard ratios (HRs) between groups. When comparing HRs between two extreme risk groups (high vs. low risk), the dataset specific cutoff that we proposed resulted in larger HRs than splitting samples into 3 groups of equal size (Supplementary Table 6).

Reviewer Figure 1. Survival differences among patients grouped by IVS. Patients were evenly distributed into 3 (a) or 4 (b) groups ordered by IVS. The indolent group with the lowest IVS is marked in red and the invasive group with the highest IVS is shown in green.

Invasive vs Indolent	Local_minima		Even number of samples in each group			
			Top vs Bottom out of 4 groups		Top vs Bottom out of 3 groups	
	HR(95% CI)	LRT p-value	HR(95% CI)	LRT p-value	HR(95% CI)	LRT p-value
Shedden et al. (n=371)	2.62(1.65-4.17)	8.30E-05	2.82(1.68-4.74)	4.30E-05	2.28(1.47-3.55)	0.0002
TCGA (n=397)	2.37(1.34-4.19)	0.004	2.26(1.28-4.00)	0.004	2.05(1.27-3.31)	0.003
Okayama et al. (n=204)	9.01(2.48-32.79)	0.0002	7.57(1.71-33.56)	0.001	6.67(1.95-22.79)	0.0003
Tang et al. (n=111)	7.31(1.47-36.25)	0.007	14.09(1.83-108.43)	0.0003	5.21(1.50-18.14)	0.003
Der et al. (n=127)	5.82(2.09-16.22)	0.0003	11.12(2.54-48.80)	4.00E-05	5.99(2.24-16.01)	4.70E-05
Rousseaux et al. (n=85)	3.17(1.15-8.76)	0.02	3.74(1.20-11.65)	0.01	3.19(1.24-8.18)	0.01
Wilkerson et al. (n=62)	3.30(1.07-10.20)	0.05	1.86(0.62-5.55)	0.26	1.61(0.62-4.16)	0.32

Reviewer Table 1. Survival differences among/between groups based on different IVS-based stratification schemes. Results for local minima-based groups are reported in the main text of our manuscript. Results for tumors separated into three or four groups of equal size based on IVS are listed here (also in Supplementary Table 6). Log-rank test p-values and hazard ratios (HRs) with 95% confidence interval (CI) are assessed for all groups or the two extreme groups (invasive vs indolent).

Taken together, the results suggest that IVS signature genes show robust prognostic value in multiple independent cohorts. We added the following on page 9.

Page 9: “To assess the robustness of survival association of IVS-based classification, we used multiple parametric and non-parametric IVS cutoff schemes to classify tumors. Both fixed IVS cutoff value and fixed IVS percentile-based classifications showed segregation of patient groups with significantly different survival (Supplementary Table 6). The hazard ratios (HRs) between the high vs. low-risk groups based on the dataset specific cutoff (Methods) were generally larger than corresponding HRs based on splits into 3 groups of equal size (SFig. 3, Supplementary Table 6).”.

Point 4.

The authors have adequately addressed this question. The invasive phenotype does not appear to be KRAS dependent.

Point 5.

The authors have adequately addressed this question. The invasive phenotype does not appear to be proliferation driven.

Point 6.

The authors partially address the reviewer’s concern. They showed that the invasion/migration phenotype occurred within 2 days prior to the proliferation slowdown of AURKA and AURKB KO cells. It is nevertheless surprising that these stable AURKA and AURKB CRISPR KO cells are viable at all and can continue to proliferate. Unless the CRISPR KO cells are actually hemizygous KO, it would be far more interesting to understand how these KO cells can transit through mitosis in the absence of AURKA or AURKB without massive chromosomal instability.

In the CRISPR/Cas9 system we used PLKO.1/GFP plasmid for sgRNA cloning. There is no antibiotic selection in this system for guide RNA carrying plasmid. The immunoblot in Figure 4 shows residual protein expression indicating persistence of AURKA and AURKB protein. We agree that the cellular biology of complete knockout of AURKA and AURKB and its impact on mitosis and chromosomal instability is of interest, however it is beyond the scope of this article.

Point 7.

The authors have done experiments to address the reviewer's concern. But the migration vs. proliferation question remains.

The reviewer acknowledges in Points 4 and 5 that the invasive phenotype does not appear to be proliferation driven, nor KRAS dependent.

Point 8.

The authors have partially addressed the reviewer's question. The plot has n=2 so it is not clear if the difference is statistically significant.

The analysis was a comparison of randomly selected regions within animals with matching tumor burden, as suggested by Reviewer 1. We agree that the biological significance is limited by the restriction to 2 animals per group that fit these criteria. We have modified the text on *page 18* as suggested by Reviewer 4.

Page 18: "Because total tumor size is associated with proliferation, we performed analysis of invasion extent in animals bearing similar tumor burden. Histopathological data from animals in the vehicle and AMG900 treatment group showed a reduction in invasive tumor with aurora kinase inhibition, further supporting a direct link between aurora kinase inhibition and tumor invasion phenotype independent of total tumor size (SFig. 11e)."

REVIEWER COMMENTS

Reviewer #4 (Remarks to the Author):

The authors addressed point 3 with a revised survival analysis. The author really didn't address point 6 in a satisfactory way, and appeared to confuse the effect of CRISPR with that of shRNA. In any given cell, CRISPR can only give a binary outcome at the allelic level: the allele would either be mutated or remain wildtype. Thus, in an unselected population after CRISPR plasmid transfection/transduction, the reduced protein expression at the population level results from a mixture of individual homozygous KO (i.e. null) cells and WT cells (including heterozygous KO, assuming neither gene is haplo-insufficient), not from all cells having a similar but lower (i.e. hypomorphic) level of protein expression. Thus, the proliferation of the KO cells in longer assays such as those in Supp Figure 8d, could be due to the loss of KO cells in the unselected population. Since the authors declined to demonstrate the long-term viability of AurA and AurB KO cells, it is difficult to separate the role of these proteins in mitosis and in metastasis beyond the first 2 days. This is important because if the KO cells are unviable, then it is meaningless to test their metastatic potential. This same problem confounds the inhibitor experiments *in vivo*, as it is difficult to address metastasis when the overall proliferation rate of the primary tumor was also decreased. There have been a number of previous studies documenting a role for these two kinases in cell migration, invasion and metastasis. For examples, PMID: 17974987, PMID: 21045147, PMID: 30082913, PMID: 22875938, PMID: 24520245, PMID: 25120775, PMID: 31996785, PMID: 33292257. Many of these prior works are not cited, and some of these prior studies point to a mitosis-independent role of AurA and AurB kinases in the regulation of actin cytoskeleton and other pathways involved in cell motility. Although this current work lends support to the aforementioned papers, it did not offer additional mechanistic insights into this process, nor does it demonstrate clearly that the metastatic effect is independent from the proliferative effect of these two kinases.

NCOMMS-21-12625C. Response to Reviewer

1. *“The authors addressed point 3 with a revised survival analysis. The author really didn’t address point 6 in a satisfactory way, and appeared to confuse the effect of CRISPR with that of shRNA. In any given cell, CRISPR can only give a binary outcome at the allelic level: the allele would either be mutated or remain wildtype. Thus, in an unselected population after CRISPR plasmid transfection/transduction, the reduced protein expression at the population level result from a mixture of individual homozygous KO (i.e. null) cells and WT cells (including heterozygous KO, assuming neither genes are haplo-insufficient), not from all cells having a similar but lower (i.e. hypomorphic) level of protein expression. Thus, the proliferation of the KO cells in longer assays such as those in Supp Figure 8d, could be due to the loss of KO cells in the unselected population. Since the authors declined to demonstrate the long-term viability of AurA and AurB KO cells, it is difficult to separate the role of these proteins in mitosis and in metastasis beyond the first 2 days. This is important because if the KO cells are unviable, then it is meaningless to test their metastatic potential”.*

With regard to the reviewer’s comment on **heterogeneity in unselected cells following CRISPR plasmid transfection**, we have a theoretical estimation as follows: Assuming a 100% infection rate, in a single CRISPR system, there are two types of resulted alleles: frameshift indels (66.7%) and in-frame indels (33.3%). Thus, in diploid cells, only 44% (0.667×0.667) of cells are homozygous for frameshift indels (no protein) while other cells carry at least one allele in-frame indels (protein with residual function) (Figure 1A, Herman et al. 2021 Biorxiv, <https://doi.org/10.1101/2021.05.20.445000>). In a 2-gene CRISPR system, it is more heterogeneous. Only 20% of cells are expected to carry homozygous frameshift indels (no protein) for both genes while the rest of cells carry alleles with in-frame indels for at least one gene.

To address the reviewer’s comments on **cell viability**, we performed experiments and show experimental evidence that transfected cells are viable at 8 days and that cells evaluated in the migration and invasion assays at day 2 are transfected and viable.

We examined long term viability in AURKA or AURKB or both CRISPR transfected H1792 cells that confirm viability at day 8. Immunoblots show residual protein expression at day 8 (Reviewer Figure 1a) and Alamar Blue assay indicates that the cells are viable (Reviewer Figure 1b). Fluorescent imaging confirms these findings for in vitro cells at day 2 and 8. (Reviewer Figure 1c, d).

Reviewer Figure1. H1792 cells CRISPR transfected with sgAURKA or sgAURKB or both. (a) Western blot for indicated protein at Day 2 and Day 8 on H1792 cells transfected with indicated sgRNA. (b) Viability assay on H1792 cells treated with indicated sgRNA using alamar blue assay.

H1792 Day 2

Reviewer Figure 1c. Bright field, GFP and mCherry imaging for H1792 cells treated with indicated sgRNA at early time point Day 2.

Reviewer Figure 1d. Bright field, GFP and mCherry imaging for H1792 cells treated with indicated sgRNA at later time point Day 8.

To confirm that the cells analyzed in the migration and invasion assays represent the impact of Aurora Kinase Knockout, we show fluorescent imaging that confirm transfection in migratory and invasive cells (Reviewer Figure 1e,f).

Reviewer Figure 1e. Bright field, GFP and mCherry imaging for H1792 cells treated with indicated sgRNA in transwell migration assay at 48hr.

H1792 Invasion Day 2

Reviewer Figure 1f. Bright field, GFP and mCherry imaging for H1792 cells treated with indicated sgRNA in transwell invasion assay at 48hr.

Fluorescence imaging data in Reviewer Fig 1e and f show cells showing reduction in migration and invasion ability are knocked out for AURKA and AURKB (shown in last bottom panel), since all cells from bright field imaging show mCherry and GFP expression for sgAURKA and sgAURKB respectively.

2. *"This same problem confounds the inhibitor experiments in vivo, as it is difficult to address metastasis when the overall proliferation rate of the primary tumor was also decreased.*

There have been a number of previous studies documenting a role these two kinases in cell migration, invasion and metastasis. For examples, PMID: 17974987, PMID: 21045147, PMID: 30082913, PMID: 22875938, PMID: 24520245, PMID: 25120775, PMID: 31996785, PMID: 33292257. Many of these prior works are not cited, and some of these prior studies point to a mitosis-independent role of AurA and AurB kinases in the regulation of actin cytoskeleton and other pathways involved in cell motility. Although this current work lends support to the aforementioned papers, it did not offer additional mechanistic insights into this process, nor does it demonstrate clearly the metastatic effect is independent from the proliferative effect of these two kinases.”

The reviewer is concerned that it is difficult to address metastasis when tumor proliferation rate is decreased by treatment. We agree that this is difficult and that this issue is relevant to understanding the mechanisms of action for numerous cancer therapeutics. From the translational science point of view, for Aurora Kinase inhibition, we do show that at 48 hours, the effect of Aurora Kinase knockout on tumor progression activities of invasion and migration is independent of cell proliferation (Supplemental Figure 7). We show *in vivo* data (Supplementary Figure 11) that demonstrates a reduction in tumor invasion, adjusted for tumor size, in animals treated with Aurora Kinase inhibitor at 13 weeks.

From the basic science point of view, we agree with the reviewer that mechanistic insights into how AURKA and AURKB interact together to affect cancer cell invasiveness are important and our group is examining this in ongoing projects. We are working with collaborators to perform CRISPR tiling screens to identify regions of AURKA, AURKB, and other 50 or so genes which are critical for cell proliferation (Herman et al, 2021, Biorxiv <https://doi.org/10.1101/2021.05.20.445000>). We are also developing CRISPR tiling screens with cell migration and invasion as reporter assays to identify regions of AURKA and AURKB critical for cell invasiveness. Then, we can determine whether AURKA's and AURKB's critical regions for cell proliferation and cell invasiveness overlap or not. These ongoing projects are beyond the current scope of translational research reported in this article.

We thank the reviewer for suggesting several related papers, some of which are cited in the article. We agree with the reviewer that these references support the role of Aurora Kinase pathway *mitosis-independent signaling* in mediating advanced stage tumor progression and metastasis. Our study focus on early stage cancer progression is novel as has been noted by the other reviewers. Another point to highlight is that knockdown both AURKA and AURKB is required for impacting lung cancer cell invasiveness, which is different from single knockdown AURKA or AURKB reported in the literature.

REVIEWERS' COMMENTS

Reviewer #4 (Remarks to the Author):

The authors provided additional evidence in their rebuttal that AurA and AurB KO cells are viable in culture over 8 days. This observation is important for supporting their conclusion, and the data in the rebuttal should be included in a supplemental figure of the paper.

NCOMMS-21-12625D. Response to Reviewer #4

1. *“The authors provided additional evidence in their rebuttal that AurA and AurB KO cells are viable in culture over 8 days. This observation is important for supporting their conclusion, and the data in the rebuttal should be included in a supplemental figure of the paper.”.*

We appreciate the positive feedback for our response for the long-term viability in AURKA or AURKB KO cells. Following the reviewer’s suggestion, we added the figures in the previous response to comments letter into SFig. 8e-h.